# Stress-induced tyrosine phosphorylation of RtcB modulates IRE1 activity and signaling outputs

Alexandra Papaioannou[1,2,*], Federica Centonze[1,2,*], Alice Metais[1,2], Marion Maurel[1,2], Luc Negroni[3,4,5], Matías Gonzalez-Quiroz[1,6,7,8], Sayyed Jalil Mahdizadeh[9], Gabriella Svensson[9], Ensieh Zare[9], Alice Blondel[1,2], Albert C Koong[10], Claudio Hetz[6,7,8], Rémy Pedeux[1,2], Michel L Tremblay[11,12], Leif A Eriksson[9], Eric Chevet[1,2]

**ER stress is mediated by three sensors and the most evolutionary conserved IRE1α signals through its cytosolic kinase and endoribonuclease (RNase) activities. IRE1α RNase activity can either catalyze the initial step of XBP1 mRNA unconventional splicing or degrade a number of RNAs through regulated IRE1-dependent decay. Until now, the biochemical and biological outputs of IRE1α RNase activity have been well documented; however, the precise mechanisms controlling whether IRE1α signaling is adaptive or pro-death (terminal) remain unclear. We investigated those mechanisms and hypothesized that XBP1 mRNA splicing and regulated IRE1-dependent decay activity could be co-regulated by the IRE1α RNase regulatory network. We identified that RtcB, the tRNA ligase responsible for XBP1 mRNA splicing, is tyrosine-phosphorylated by c-Abl and dephosphorylated by PTP1B. Moreover, we show that the phosphorylation of RtcB at Y306 perturbs RtcB interaction with IRE1α, thereby attenuating XBP1 mRNA splicing. Our results demonstrate that the IRE1α RNase regulatory network is dynamically fine-tuned by tyrosine kinases and phosphatases upon various stresses and that the extent of RtcB tyrosine phosphorylation determines cell adaptive or death outputs.**

## Introduction

The imbalance between the cellular demand to fold secretory and transmembrane proteins and the ER capacity to achieve this function can result in the accumulation of improperly folded proteins in this compartment, a situation known as ER stress (Almanza et al, 2018). The activation of the ER stress sensors inositol-requiring enzyme 1 alpha (IRE1), activating transcription factor 6 alpha (ATF6α), and protein kinase RNA (PKR)–like ER kinase (PERK) aims to restore ER homeostasis and is known as the adaptive unfolded protein response (UPR). However, when the stress cannot be resolved, the UPR triggers cell death (McGrath et al, 2021), which is referred to as terminal UPR. Thus far, the mechanisms controlling the switch between adaptive (aUPR) and terminal (tUPR) UPR remain incompletely characterized. Among the possible candidates, the IRE1 pathway, being greatly conserved through evolution, plays crucial roles in both physiological and pathological ER stress. Similar to the other sensors, IRE1 is an ER transmembrane protein activated after dissociation from binding immunoglobin protein and/or direct binding to improperly folded proteins (Karagöz et al, 2017).

IRE1 is characterized by the presence of both kinase and endoribonuclease domains in its cytosolic region. After its dimerization/oligomerization, IRE1 trans-autophosphorylates, allowing the recruitment of TRAF2 and subsequent activation of the JNK pathway (Urano et al, 2000). IRE1 dimerization and phosphorylation also yield a conformational change, which activates its RNase domain and leads to unconventional splicing of the XBP1 mRNA and subsequent expression of a major UPR transcription factor XBP1s (Calfon et al, 2002; Lee et al, 2002). Importantly, the ligation following the IRE1-mediated cleavage of the 26-nucleotide intron in the XBP1 mRNA is catalyzed by the tRNA ligase RtcB (Jurkin et al, 2014; Kosmaczewski et al, 2014; Lu et al, 2014; Ray et al, 2014). The IRE1 RNase domain also promotes the cleavage of mRNA (Hollien & Weissman, 2006; Hollien et al, 2009), rRNA (Iwawaki et al, 2001), and miRNA (Lerner et al, 2012; Upton et al, 2012) sequences, a process named regulated IRE1-dependent decay (RIDD) of RNA (Hollien et al, 2009). Interestingly, the degree of oligomerization of IRE1α may define its RNase activity toward XBP1 mRNA splicing or RIDD, ultimately impacting on cell fate (Upton et al, 2012; Le Thomas et al, 2021 Preprint; Li et al, 2021). Although recent and elegant direct

[1]INSERM U1242, University of Rennes, Rennes, France    [2]Centre Eugène Marquis, Rennes, France    [3]Centre National de la Recherche Scientifique, UMR7104, Illkirch, France    [4]Institut National de la Santé et de la Recherche Médicale, U1258, Illkirch, France    [5]Université de Strasbourg, Illkirch, France    [6]Biomedical Neuroscience Institute, Faculty of Medicine, University of Chile, Santiago, Chile    [7]Center for Geroscience, Brain Health and Metabolism (GERO), Santiago, Chile    [8]Program of Cellular and Molecular Biology, Institute of Biomedical Sciences, University of Chile, Santiago, Chile    [9]Department of Chemistry and Molecular Biology, University of Gothenburg, Göteborg, Sweden    [10]Department of Radiation Oncology, The University of Texas MD Anderson Cancer Center, Houston, TX, USA    [11]Goodman Cancer Research Centre, McGill University, Montreal, Canada    [12]Department of Biochemistry, McGill University, Montreal, Canada

Correspondence: eric.chevet@inserm.fr; leif.eriksson@chem.gu.se
*Alexandra Papaioannou and Federica Centonze contributed equally to this work.

approaches were used to measure the oligomeric changes that underpin IRE1 activation (Belyy et al, 2021 *Preprint*), it remains unclear which is the order of oligomerization required for *XBP1* mRNA splicing versus RIDD activity. However, there is a consensus on the cytoprotective effects of XBP1s and the cell death–inducing outputs of RIDD under acute ER stress (Han et al, 2009; Upton et al, 2012; Tam et al, 2014). RIDD was also described to maintain ER homeostasis under basal conditions (Maurel et al, 2014). In this model, whereas *XBP1* mRNA splicing is induced during the adaptive UPR and inactivated during the terminal UPR, RIDD displays an incremental activation pattern reaching unspecific RNA degradation during tUPR. Despite our knowledge on IRE1 signaling biological outputs, little is known about the integration of these two RNase signals and how their balance impacts on the life and death decisions of the cell. The existence of a multiprotein complex recruited in IRE1 foci that dynamically changes composition during the course of ER stress, named "UPRosome," has been suggested as a possible modulation mechanism (Hetz & Glimcher, 2009; Hetz & Papa, 2018). This is supported by the recent identification of IRE1 interactors (e.g., PP2A and TUBα1a) regulating XBP1 mRNA splicing different from RIDD (Sepulveda et al, 2018).

Herein, we hypothesized that *XBP1* mRNA splicing and RIDD are part of an autoregulatory IRE1 RNase network. RIDD targets could possibly impact on the splicing of *XBP1* mRNA, thus fine-tuning the response to ER stress and impacting on subsequent cell fate decisions. We identified *PTP1B* mRNA as a RIDD target affecting XBP1s activity by dephosphorylating RtcB, the tRNA ligase that performs *XBP1s* ligation. The tyrosine kinase c-ABL is also part of this network as it phosphorylates RtcB, thus revealing a posttranslational regulatory mechanism of the IRE1 RNase signaling outputs. The study furthermore supports the true existence and dynamic nature of a UPRosome whose composition can determine the outcome of IRE1 signaling.

## Results

### PTP1B contributes to *XBP1* mRNA splicing and is a RIDD target

We first hypothesized that molecules involved in the IRE1-XBP1 pathway could at the same time be RIDD target and as such could be part of a regulatory mechanism between both activities. To identify XBP1s regulators in the context of ER stress and possible RIDD substrates, we conducted two independent screens. First, an siRNA library against ER protein–coding RNAs was used to transfect HEK293T cells expressing an XBP1s-luciferase reporter (Spiotto et al, 2010) (Fig S1A). The cells were then subjected to tunicamycin (Tun)-induced ER stress, and luciferase activity was monitored to identify XBP1s-positive and XBP1s-negative regulators (Fig 1A). We identified 23 positive and 32 negative regulators (out of an siRNA library targeting >300 genes) including candidates involved in metabolic processes (e.g., CH25H and DHCR7), posttranslational modifications (e.g., POMT2 and DMPK), ERAD pathway (e.g., SYVN1 and UBC6), and cell growth (e.g., RRAS and CD74) (Fig 1B). Second, to identify mRNA that could be cleaved by IRE1, we performed an in vitro IRE1 mRNA

cleavage assay as described previously (Lhomond et al, 2018) (Fig S1B), yielding a total of 1,141 potential RIDD targets. XBP1s regulators that could also be subjected to RIDD-mediated degradation were determined by intersecting the hit lists from both screens, yielding seven candidates, namely, ANXA6, ITPR3, PTPN1 (positive XBP1s regulators), RYR2, TMED10, KTN1, and ITPR2 (negative XBP1s regulators). The lists from this study were later combined with a list of XBP1s regulators obtained from a genome-wide siRNA screen (Yang et al, 2018). Although we did not observe any common XBP1s modulators with our own study, likely explained by a sensitivity issue due to the targeted aspect of the library used in the present work, we note the existence of 33 more hits in the intersection between XBP1 splicing and RIDD. The functional association network of the reported candidates showed their involvement in UPR regulation, RNA processing, and cellular homeostasis (Figs. 1C and D and S1C). Interestingly, among the seven hits found was PTPN1, which encodes for the protein tyrosine phosphatase 1B (PTP1B), a protein previously reported by us to potentiate XBP1 splicing in response to ER stress (Gu et al, 2004).

To confirm our initial results, we repeated these experiments by using RT-qPCR. PTP1B$^{-/-}$ (KO) MEFs exhibited a significantly lower capacity to yield *XBP1* mRNA splicing under Tun-induced ER stress than their PTP1B$^{+/+}$ (WT) counterparts (four- to fivefold increase in PTP1B KO cells versus sevenfold increase in PTP1B WT cells; Figs 1E and S1D). Regarding the UPR sensors ATF6 and PERK, the expression of both binding immunoglobin protein and HERPUD was also attenuated in PTP1B$^{-/-}$ cells, whereas that of CHOP was increased, most likely as a compensatory mechanism (Fig S1E–G). Using the same cellular settings, we also observed the basal activity of all three UPR sensors (Fig S1H). We next sought to test whether PTP1B mRNA was indeed a RIDD substrate. To this end, we used U87 cells expressing a dominant negative form of IRE1α (IRE1 DN) or an empty vector (Drogat et al, 2007). We conducted an actinomycin D chase experiment under basal or ER stress conditions to evaluate the posttranscriptional regulation of PTP1B mRNA expression. Under basal conditions, U87 DN cells showed higher expression levels of PTP1B than the control cells (Fig 1F). Moreover, upon ER stress (DTT or Tun), PTP1B mRNA degradation was more efficient in control cells than in DN cells (Figs 1G and S1I). This was a phenomenon also observable at the level PTP1B protein levels, which significantly decreased upon ER stress compared with control cells (Fig S1J). Collectively, these results show that PTP1B mRNA is a genuine RIDD target and that PTP1B promotes IRE1-dependent XBP1 mRNA splicing.

### RtcB is tyrosine-phosphorylated by c-ABL and dephosphorylated by PTP1B

Based on the experiments reported previously, we concluded that PTP1B may represent a key regulator of IRE1 activity and investigated how this protein could mechanistically alter IRE1 signaling. PTP1B is a tyrosine phosphatase, and thus, we first searched the PhosphoSitePlus database (Hornbeck et al, 2015) for the presence of phosphotyrosine (pY) residues in proteins described previously to regulate IRE1/XBP1s signaling. This revealed that although there was no reported pY residue for IRE1 (a result confirmed experimentally in our laboratory), six pY residues were reported for RtcB

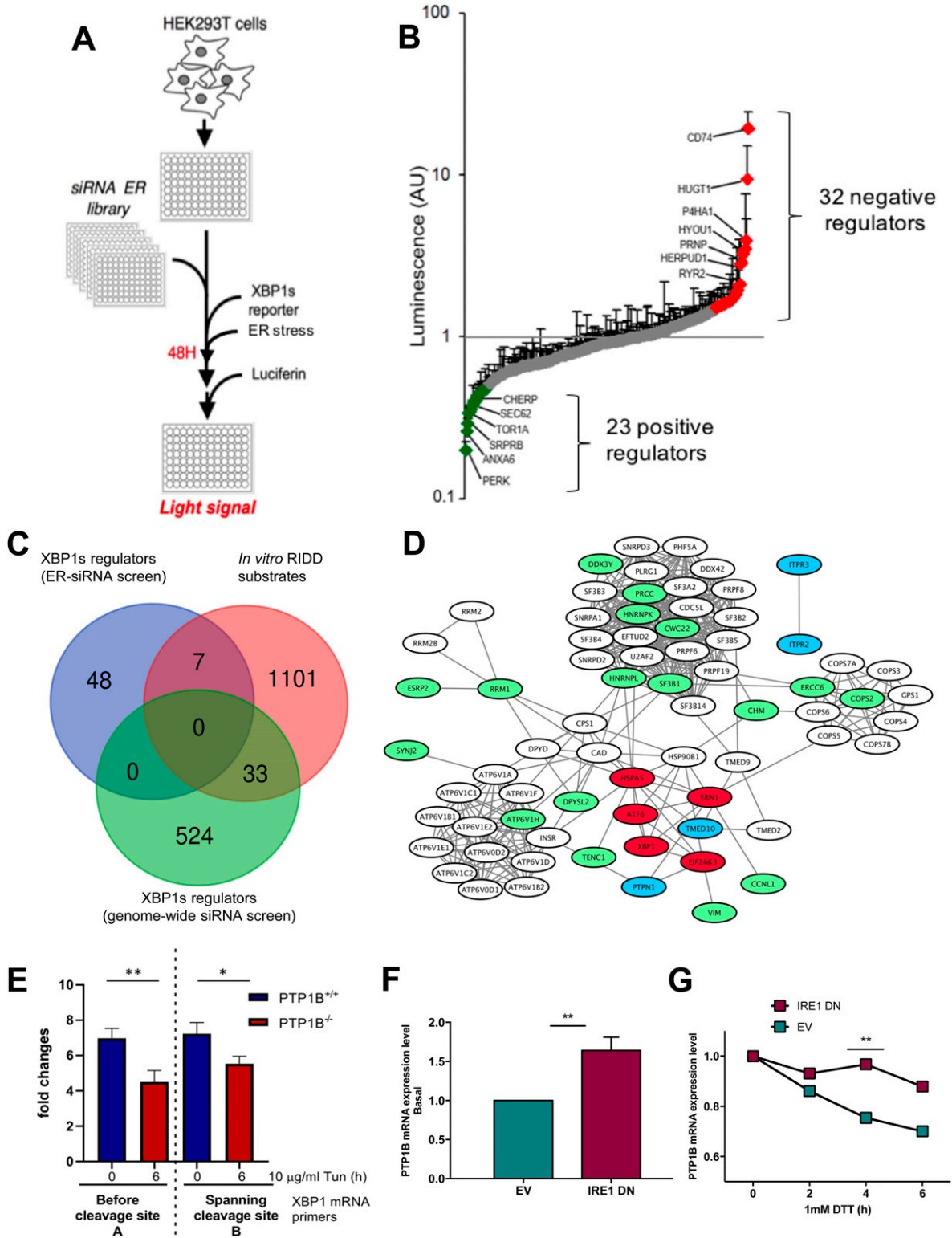

**Figure 1.  SiRNA screening and in vitro IRE1α cleavage assay identify PTP1B as both XBP1 splicing regulator and regulated IRE1-dependent decay target.**
**(A)** siRNA-based screening assay: HEK293T cells were transfected with siRNA sequences against genes encoding ER proteins. They were subsequently transfected with an XBP1s-luciferase reporter (Fig S1A), and after 48 h and the induction of ER stress, the cells were tested for the intensity of light signal after the addition of luciferin. **(A, B)** Luminescence quantification of the screening assay described in (A). **(A, B, C)** Venn diagram of the gene list resulting from the assay described in (A, B), a list of possible regulated IRE1-dependent decay substrates obtained from an in vitro IRE1α cleavage assay (Fig S1B) and a genome-wide siRNA-based screening assay (described in Yang et al [2018]). **(C, D)** A schematic network of the genes in the intersections of the lists as described in (C). **(E)** PTP1B$^{+/+}$ (WT) and PTP1B$^{-/-}$ (KO) MEFs were

and found in phosphoproteomics studies (Table S1, [Hornbeck et al, 2015; Tsai et al, 2015; Bian et al, 2016]). Interestingly, multiple protein sequence alignment of the human RtcB and its orthologs in different species from the metazoan/animal kingdom revealed not only a high conservation of the protein across evolution but also the conservation of specific tyrosine residues (Fig S2A). To monitor RtcB tyrosine phosphorylation in cellular models, we transfected HEK293T cells with the pCMV3-mouse RTCB-Flag plasmid and treated them or not with the tyrosine phosphatase inhibitor bpV(phen). Cell extracts were immunoprecipitated with anti-Flag antibodies, and the immune complexes were immunoblotted using anti-pY antibodies. BpV(phen) treatment augmented the pY signal in the whole-cell lysates (Fig S2B) and showed that endogenous RtcB was also subjected to tyrosine phosphorylation (Fig S2C). We are aware that bp(V)phen treatment might create a bias in the analysis, but in this case, systemic inhibition of protein tyrosine phosphatases was a prerequisite for the visualization of pY-RtcB.

We then tested whether RtcB interacts with PTP1B. To this end, HEK293T cells were transfected with RtcB-Flag and either PTP1B-WT or PTP1B-C215S (trapping mutant; [Jia et al, 1995; Zhang et al, 2000]) expression plasmids. Cell extracts were immunoprecipitated using anti-PTP1B antibodies, and the immune protein complexes were immunoblotted using anti-RtcB antibodies. RtcB-Flag exhibited a more stable association with the C215S PTP1B than that observed for its wild-type form (Fig 2A), suggesting that the pY residue(s) in RtcB might be trapped by the mutant PTP1B. To further document the functional relationship between PTP1B and RtcB tyrosine phosphorylation, mRtcB-Flag was transfected with PTP1B$^{-/-}$ and PTP1B$^{+/+}$ MEFs, and the RtcB-Flag pY status was evaluated as described previously. This revealed that the mRtcB-Flag pY signal was increased in PTP1B$^{-/-}$ MEFs when compared with PTP1B$^{+/+}$ MEFs (Fig 2B), which demonstrates that PTP1B is involved in the dephosphorylation of RtcB.

We next sought to identify a tyrosine kinase that could be responsible for RtcB phosphorylation. It was shown that the tyrosine kinase c-ABL acts as a scaffold for IRE1 by assisting its oligomerization and thus enhancing terminal UPR but is not directly involved in IRE1 phosphorylation (Morita et al, 2017). To document the possible phosphorylation of RtcB by c-ABL, we performed an in vitro kinase assay using recombinant human His-tagged c-ABL as kinase and recombinant human GST-RtcB as the substrate. This experiment showed that c-ABL phosphorylated RtcB in vitro (Fig S2D). We then mapped the tyrosine phosphorylation sites on RtcB using mass spectrometry and identified three tyrosine residues Y306, Y316, and Y475 that were subjected to phosphorylation (Fig 2C). These tyrosines were also reported in independent proteomics screens to be phosphorylated (Table S1). Among them, the LVMEEAPESpY$^{475}$K phosphopeptide was found not only in the purified TiO$_2$ but also in the crude sample, indicating its probable greater abundance. In an attempt to use this phosphopeptide as a proxy of RtcB tyrosine phosphorylation, we developed an immunological tool for the selective detection of pY of RtcB at Y475, as shown in Fig S3A–C. We observed the significantly reduced mRtcB-Flag tyrosine phosphorylation when cells were treated with the tyrosine kinase inhibitor dasatinib, which targets c-ABL in a specific manner and which may also affect the activity of other tyrosine kinases such as SRC (Fig 2D and E). Finally, we tested how RtcB tyrosine phosphorylation was affected upon ER stress and in c-ABL–depleted cells (Fig 2F and G). These experiments revealed that global tyrosine phosphorylation of RtcB-WT increased upon tunicamycin (TM) treatment and decreased in c-ABL–knocked down cells (Fig 2F and G). This supports our finding that pY-mediated regulation of RtcB is an ER stress–related event.

## Molecular dynamics (MD) simulations of tyrosine-phosphorylated RtcB and possible outcomes

We next evaluated the impact of tyrosine phosphorylation on the RtcB structure using MD simulations. The starting structure was a recent homology model of the human RtcB protein (Nandy et al, 2017). The systems that were prepared and subjected to MD simulations were distinguished by the presence (active) or absence (inactive) of bound guanosine monophosphate (GMP) at the active site of RtcB and by the different combinations of pY. After the simulations, certain conformational differences on the RtcB systems such as the movement of helices or specific amino acids were observed (Figs 3A and B, S4A and B, and S5A and B). By itself, the addition of a phosphoryl group to a tyrosine gives a negative charge to this modified amino acid, which in turn induces local changes to both the electrostatic surface and the conformation. Collectively, these alterations can have implications on the protein structure, substrate specificity, and interaction surface. To quantify these changes, we performed post-MD analyses including determination of the solvent accessibility surface area (SASA) and pKa values for the three tyrosine residues, as well as characterization of each tyrosine residue as "buried" within the protein structure or more solvent exposed (Tables S2 and S3). In addition, we quantified the dihedral angles for each of the three tyrosines in all the different RtcB systems as a measure of their rotation (Fig 3C). We observed that when RtcB is fully phosphorylated, Y306 completely changes orientation to become more solvent exposed than buried in the body of the ligase (Fig 3D, A, and B). pY306 is characterized as a residue existing in both fully buried and solvent-exposed conformations (Table S2). Consistently, analysis of the final structures after the 200-ns simulations revealed that although the differences in the SASA values are not striking, pY306 has the highest value of SASA among the three pYs (pY306/pY316/pY475) and a higher SASA value than Y306 in the non-pY system (Table S3). Next, we measured the docking score of GTP in the active site of RtcB in the

treated for 0 and 6 h with 10 μg/ml TM. RNA was isolated, and the resulting samples were analyzed with qPCR for spliced and total *XBP1* mRNA levels. The bar graph presents the tunicamycin-induced fold change in *XBP1* mRNA splicing between PTP1B$^{+/+}$ (WT, blue) and PTP1B$^{-/-}$ (KO, red). Data information: Data values are the mean ± SEM of four independent experiments. The unpaired *t* test was applied for the statistical analyses comparing the values (mean ± SEM) for the 0 and 6 h time points. **P = 0.0074; *P = 0.0187. Total *XBP1* mRNA qPCR primers amplifying the region before the cleavage site or the region spanning the cleavage site were used (Fig S1C). **(F, G)** PTP1B mRNA expression levels in U87 cells expressing or not a dominant negative form of IRE1α at 0 h time point (F) and during DTT treatment (1 mM) with a 2-h Actinomycin D (5 μg/ml) pretreatment (G). EV, empty vector; IRE1 DN, dominant negative (cytosolic-deficient) form of IRE1α.

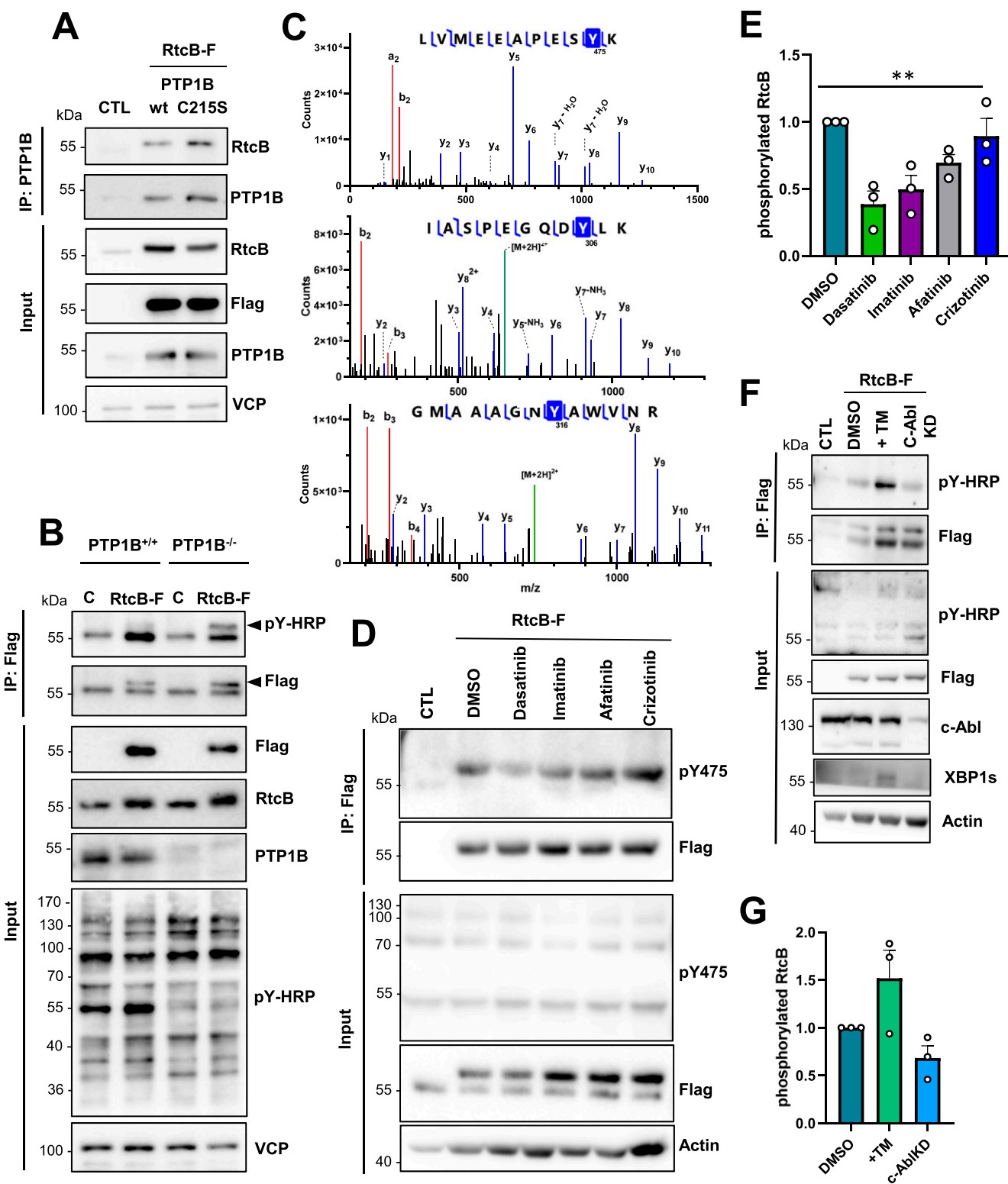

**Figure 2. RtcB is a substrate of the tyrosine kinase c-Abl and the tyrosine phosphatase PTP1B.**
**(A)** HEK293T cells were left non-transfected (CTL) or transfected with 1 µg of the wt RtcB-Flag and 1 µg of either the wt PTP1B plasmid or C215S mutant one. Immunoprecipitation (IP) was carried out in the cell lysates with the PTP1B antibody, the immunoprecipitates were immunoblotted for RtcB, and the membrane was re-probed with PTP1B. Inputs probed for RtcB, Flag, PTP1B, and VCP are shown. **(B)** PTP1B$^{+/+}$ or PTP1B$^{-/-}$ MEFs were left untransfected (CTL) or transfected with 2 µg wt RtcB-Flag and treated with 15 µM bpV(phen) for 2 h. IP was performed in the cell lysates using Flag Ab, the immunoprecipitates were immunoblotted for phosphotyrosine, and the membrane was re-probed with Flag. Input samples probed for Flag, RtcB, PTP1B, pY-HRP, and VCP are shown. Black arrowheads indicate the RtcB-Flag protein, and

differentially tyrosine-phosphorylated systems. The phosphorylation of Y475 totally abrogated the binding of GTP in the active site (Fig 3E and F). This is not surprising because Y475 is located at the active site of the protein and the bulky phosphoryl group creates a barrier that does not allow GTP to bind to the His 428 residue for the subsequent enzymatic catalysis.

## Cellular impact of RtcB tyrosine phosphorylation

We then tested the effects of preventing phosphorylation of these tyrosine residues in a cellular environment. To this end, we generated RtcB-Flag Y-to-F mutants (Fig 4A) using site-directed mutagenesis to keep the aromatic ring in a conformation that cannot be phosphorylated. Single (Y306F, Y316F, Y475F), double (Y306F/Y316F), and triple (Y306F/Y316F/Y475F) mutants were obtained. HEK293T cells were transfected with the WT and mutant plasmids, treated with bpV(phen), and lysates were then immunoprecipitated with anti-Flag antibodies. Anti-pY immunoblot revealed that although the Y306F and Y316F single mutants showed similar levels of tyrosine phosphorylation to WT, the Y475F mutant had a reduced degree of tyrosine phosphorylation, indicating a potential preference for this residue by c-ABL (Fig 4B). The double Y306F/Y316F mutant showed reduced tyrosine phosphorylation compared with the corresponding single mutants (Fig 4B). The triple mutant still displayed tyrosine phosphorylation, suggesting that other Tyr residues could be phosphorylated most likely by other tyrosine kinases. The quantitation of these Western blots indicated a decrease in tyrosine phosphorylation correlating with the number of tyrosine to phenylalanine conversion (Fig 4C). Moreover, we tested how tyrosine phosphorylation of the different RtcB phospho-ablating mutants was affected upon TM-induced ER stress and in c-ABL–depleted cells as previously shown in Fig 2F and G. Tyrosine phosphorylation of the RtcB mutants appeared, in general, to be less affected than that of RtcB-WT (Fig S6A). This could be an indication that the phosphorylation events on single tyrosine residues studied herein are interconnected and might regulate each other. Of note, tyrosine phosphorylation monitored for RtcB-Y306F upon TM-induced ER stress exhibited an opposite behavior to that of RtcB-WT. Tyrosine phosphorylation of the RtcB-Y475F mutant appeared to be the least affected, supporting our observation that Y475 might be the most abundant pY in RtcB. Our results also showed that in c-ABL–depleted cells, RtcB-Y306F behaved similar to RtcB-WT. In contrast, RtcB-Y316F showed an increased tyrosine phosphorylation, which was the opposite of that observed for both RtcB-WT and RtcB-Y306F. This observation might be indicative of the implication of other kinases

phosphorylating these residues outside c-ABL. Furthermore, RtcB-Y475F appeared to be the least affected when c-ABL was depleted. Again as indicated before, this result supported our initial finding that Y475 might be the best c-ABL substrate (as reflected by its abundance) in RtcB (Fig S6A). Collectively, our results confirm that RtcB contains tyrosine residue substrates of c-ABL kinase activity. Moreover, they may indicate that the sequence of tyrosine phosphorylation on RtcB might impact on the relative phosphorylation of the different tyrosines. The additional pYs found in previous mass spectrometry–based analyses are listed in Table S1.

The interaction of each RtcB mutant with the substrate-trapping mutant PTP1B-C215S was also tested as described earlier for RtcB-WT in Fig 2A. The double (Y306F/Y316F) and mostly the triple (Y306F/Y316F/Y475F) mutants showed a statistically significant reduction of their physical association with the PTP1B-trapping mutant compared with RtcB-WT (Fig S6B and C). This could be indicative of a lack of preference for PTP1B to bind to one of the tyrosine sites but rather that the tyrosine phosphatase is able to bind and dephosphorylate all three sites Y306, Y316, and Y475. Following this observation, we generated HeLa cell lines stably expressing either the wt or the mutant forms of RtcB-Flag (including also the catalytically inactive mutant H428A) described earlier (Fig 4A). Cycloheximide chase experiments were conducted on these cells to compare the protein stability of each RtcB-Flag forms. These analyses revealed that all RtcB-Flag proteins display a similar half-life (Figs 4D and E and S7A–C), which was comparable with that of endogenous RtcB (Figs 4D and E and S7D).

## Phosphorylation of RtcB at Y306 attenuates XBP1 mRNA splicing and enhances RIDD activity

Because we confirmed that RtcB could be phosphorylated in cells, we next tested the impact of these phosphorylation sites on XBP1 mRNA splicing. To this end, we treated stably expressing HeLa RtcB-Flag mutant cell lines with 1 µg/ml Tun for 0, 2, 4, 8, 16, and 24 h to monitor the XBP1 mRNA splicing (Fig 5A and D), and the results were normalized to the expression of RtcB-Flag (Fig 5C and F), thereby indicating RtcB-Flag-specific activity with respect to XBP1 mRNA splicing (Fig 5B and E). We observed that cells expressing the Y306F RtcB-Flag showed increased XBP1 mRNA splicing activity (Fig 5A and B). Furthermore, cells expressing the double mutant exhibited reduced XBP1 mRNA splicing compared with RtcB-WT-Flag but performed slightly better than cells expressing the catalytically inactive mutant H428A (Fig 5D and E). The latter matched MD

white arrowheads indicate an unspecific band at 55 kD. **(C)** Samples from a scaled up in vitro kinase reaction containing recombinant human c-ABL and RtcB were analyzed using mass spectrometry. Fragment spectra corresponding to three different phospho-peptides containing tyrosine residues are depicted (y ions are shown in blue and b ions in red). **(D)** HEK293T cells transfected with the wt RtcB-Flag were treated 24 h post-transfection with 10 µM of tyrosine kinase inhibitors afatinib, crizotininb, dasatinib, and imatinib for 8 h and 15 µM bpV(phen) for 2 h. IP was performed in the cell lysates using Flag antibody, and the immunoprecipitates were first immunoblotted for RtcB-pY475 and then re-probed with Flag Ab. Input samples probed for pY-HRP, Flag, and VCP are shown. **(D, E)** The levels of phosphorylated RtcB (D) were normalized to RtcB protein levels (D). **(F)** HEK293T cells transfected with the Flag-RtcB-WT were treated 24 h post-transfection with 1 µg/ml TM for 6 h or transfected with c-ABL siRNA for 2 d. IP was carried out in the cell lysates using Flag antibody, and the immunoprecipitates were first immunoblotted for pY and then re-probed with Flag antibodies. Input samples probed for pY-HRP, Flag, c-ABL, XBP1s, and actin are shown. **(F, G)** The levels of phosphorylated RtcB (F) were normalized to RtcB protein levels (F). Data information: The blots shown are representative of three or more independent experiments. Data shown in the graphs correspond to the mean ± SEM of n = 3 independent experiments. One-way ANOVA was applied for the statistical analyses (**$P < 0.01$).
Source data are available online for this figure.

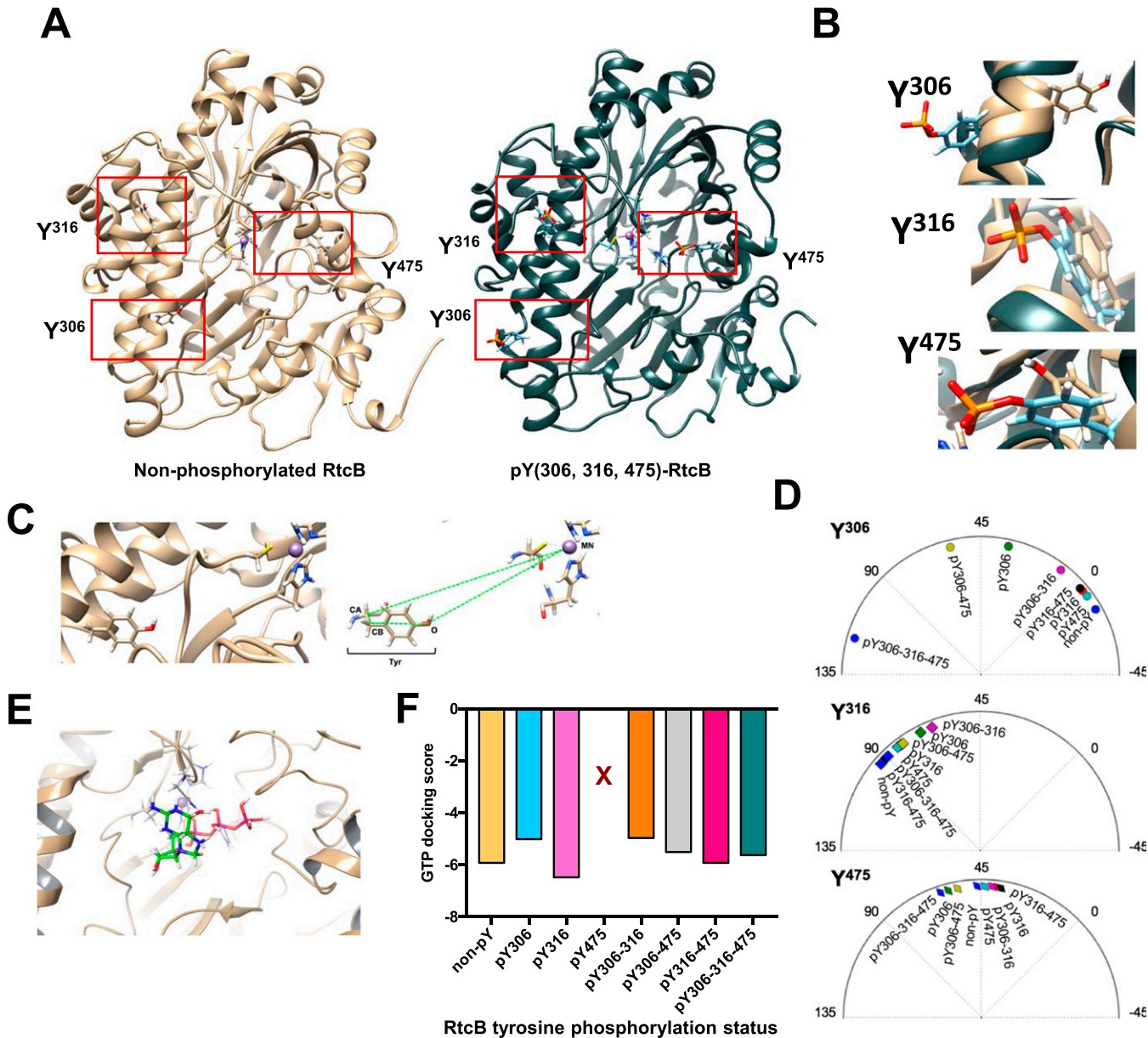

**Figure 3. Molecular dynamics (MD) of RtcB tyrosine-phosphorylated systems and possible implications.**
**(A)** Ribbon-like structures of the RtcB protein resulting from the MD simulations either non-phosphorylated (left) or phosphorylated at Y306, Y316, and Y475 (right).
**(A, B)** Zoom in at Y306, Y316, and Y475 after superposition of the two RtcB models shown in (A). **(C)** Scheme explaining the calculation of a dihedral angle for a tyrosine relative to the $Mn^{2+}$ in the active site of RtcB. The example shows Y306 in the unphosphorylated RtcB system. **(D)** The dihedral angles in degrees (°) of each of the three tyrosines 306, 316, and 475 relative to the metal ion in the protein active site in all RtcB systems after 200-ns MD simulation, depicted in polar charts. **(E)** Scheme showing the docked GTP ligand in the active site of the unphosphorylated RtcB protein. **(F)** Glide XP docking scores of GTP in the active site of the different RtcB systems. X represents the absence of GTP binding to the active site when Y475 is phosphorylated.

simulation results, indicating the that phosphorylation of RtcB on Y475 might abrogate the binding of GTP in the active site of the protein, thereby halting its catalytic activity. Interestingly, cells expressing the triple mutant showed a slightly lower XBP1 mRNA splicing than those expressing the H428A mutant, thereby suggesting that the complete absence of c-ABL–mediated pY negatively affects RtcB in its contribution to XBP1 mRNA splicing (Fig 5D and E). Because of the observed effect of Y306F RtcB on XBP1 mRNA

splicing, we wondered whether RIDD activity would change as well. To address this, we performed an actinomycin D chase experiment under ER stress (induced by either Tun or DTT) and monitored the expression PER1 and SCARA3 mRNAs, two known RIDD substrates whose expression levels decreased upon Tun-induced ER stress (Fig S7E). Upon ER stress, Y306F RtcB-Flag–expressing cells showed higher levels of PER1 and SCARA3 mRNA than control cells (Fig 5G) corresponding to lower RIDD activity. The same was observed even

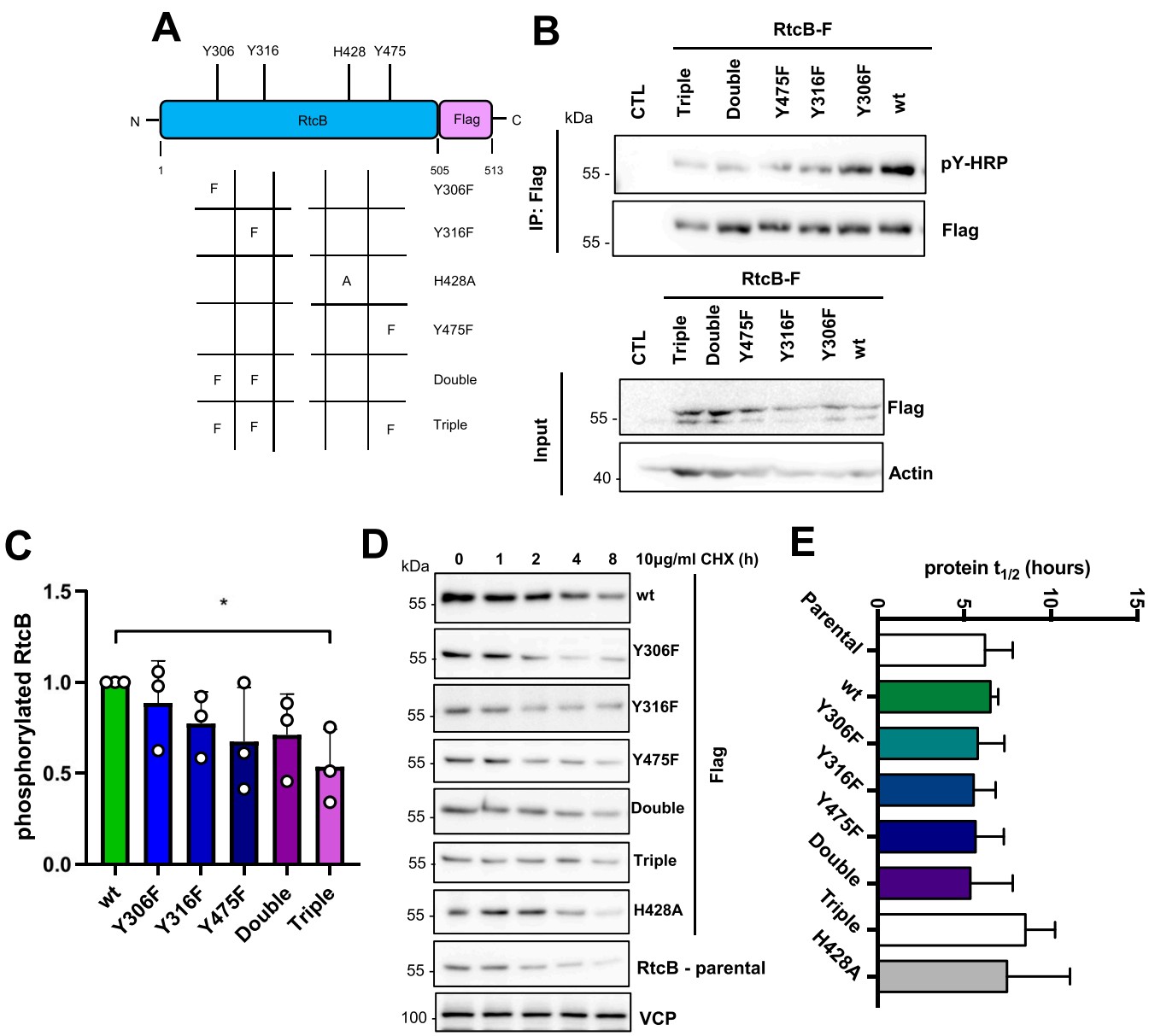

**Figure 4. Generation of cellular models with altered RtcB tyrosine phosphorylation status.**
**(A)** Scheme showing the RtcB-Flag protein encoded by the pCMV3-mRtcB-Flag plasmid and the mutant forms that were created in this study. **(B)** HEK293T cells were left untransfected (CTL) or transfected with 2 $\mu$g of the different RtcB-Flag plasmids. 24 h after transfection, the cells were treated with 15 $\mu$M bpV(phen) for 2 h. Immunoprecipitation was performed in the cell lysates using Flag antibody, and the immunoprecipitates were immunoblotted for phosphotyrosine. The membrane was re-probed with Flag. The corresponding input samples are also shown immunoblotted for Flag and actin. **(B, C)** The levels of phosphorylated RtcB (B) were normalized to RtcB protein levels (B). **(D)** HeLa stable lines expressing wt or mutant RtcB-Flag were treated with 10 $\mu$g/ml cycloheximide for 0, 1, 2, 4, and 8 h to block translation and monitor the current protein levels. Here, RtcB-Flag expression levels were monitored in the resulting protein lysates. The parental HeLa cells were included to additionally monitor the levels of the endogenous RtcB. VCP was used as a loading control. **(E)** $t_{1/2}$ calculated for the different forms of the Flag-RtcB-WT protein or the endogenous RtcB in the case of the parental HeLa cells. Data information: The blots are representative of three independent experiments. Data presented in the graphs correspond to the mean ± SEM of three independent experiments. One-way ANOVA was applied for the statistical analyses (*$P < 0.02$). Source data are available online for this figure.

when the endogenous RtcB protein was knocked down using an siRNA targeting the 3′-UTR of the RTCB mRNA (Fig 5G and H). Despite the changes in *XBP1* mRNA splicing and RIDD activity, when the splicing of the three intron-containing tRNA molecules in humans (corresponding to the Tyr, Arg, and Ile tRNA) was compared between the Y306F and the RtcB-WT-Flag–expressing cells, there was no change in tRNA splicing activity (Fig 5I). As a control to this

experiment, the levels of the intronless tRNA molecules Pro and Val were also tested (Fig 5J). These results indicated that phosphorylation of RtcB on Y306 represents a regulatory mechanism to control the balance between XBP1 mRNA splicing and RIDD activity downstream of IRE1 RNase. The specificity of this regulation is consistent with the fact that tRNA splicing is not influenced by the RtcB Y306F mutation.

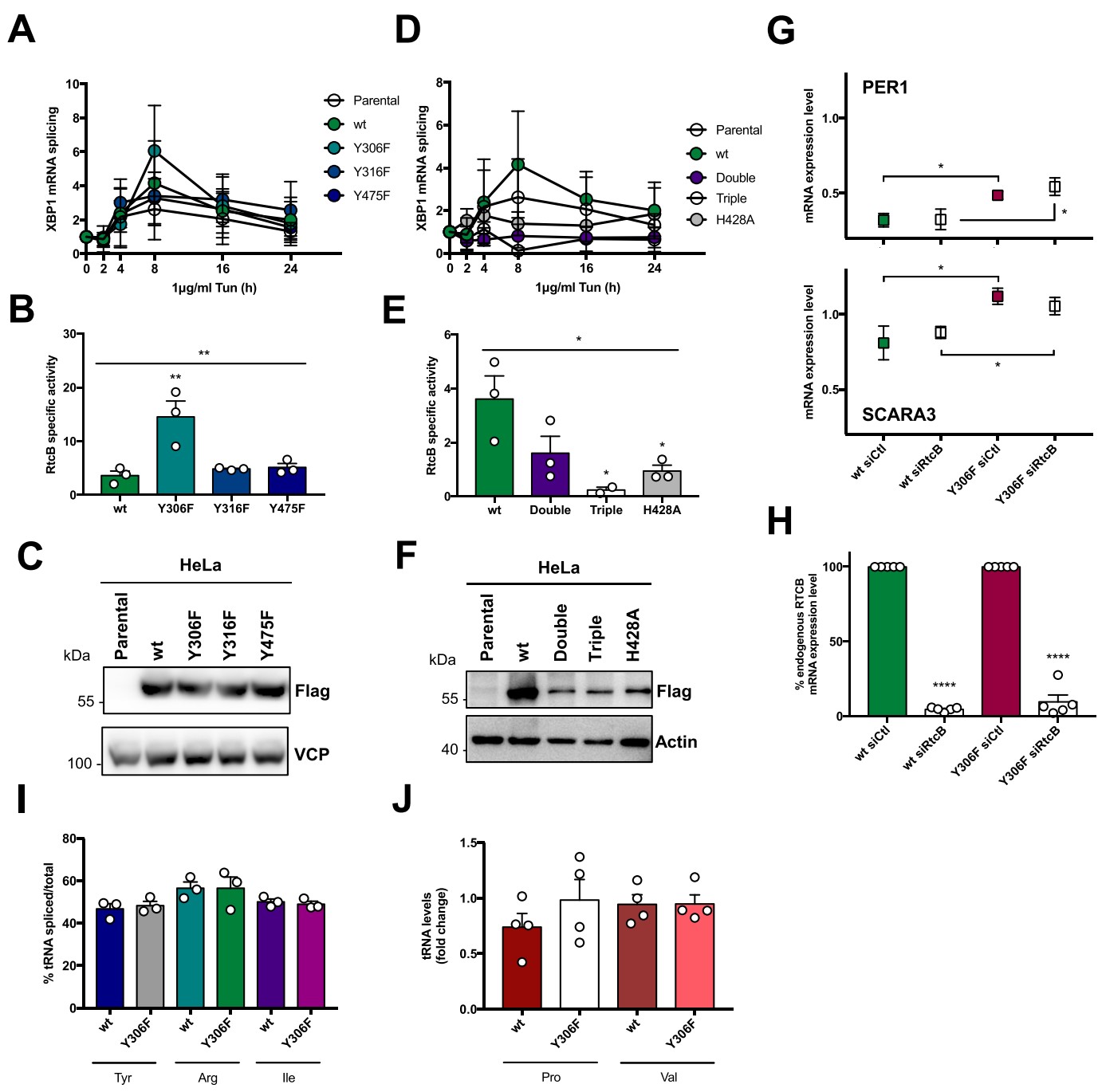

**Figure 5. Phosphorylation of RtcB on Y306 negatively affects XBP1 mRNA splicing but not regulated IRE1-dependent decay activity.**
**(A, B, C, D, E, F)** HeLa lines stably expressing wt or mutant RtcB-Flag were treated with 1 μg/ml tunicamycin for 0, 2, 4, 8, 16, and 24 h. **(A, D)** cDNA from these was analyzed by qPCR for XBP1s and XBP1 total mRNA levels, with their ratio being the XBP1 mRNA splicing as depicted in the y axis of the graphs. The parental HeLa cells are also included for comparison. **(A, B, C, D, E, F)** The peak of XBP1s activity at 8 h (A, D) was then normalized to the protein levels (C, F) to obtain the RtcB-specific activity. **(C, F)** Stable HeLa lines were lysed and analyzed for Flag-RtcB-WT expression levels by immunoblotting with anti-Flag. Actin and VCP were used as loading controls. **(G)** HeLa lines stably expressing WT or Y306F Flag-RtcB were left untransfected or transfected with an siRNA targeting the 3'-UTR of the RTCB mRNA. 48 h post-transfection, the cells were pretreated for 2 h with 5 μg/ml actinomycin D and then treated either with 5 μg/ml TM for 4 h or 1 mM DTT for 2 h. The graph shows the mRNA levels of PER1 (upper part) and SCARA3 (lower part) after normalization to the untreated samples (0 h time point). **(G, H)** RT-qPCR analysis of the untreated samples from (G) for the endogenous RTCB mRNA using primers spanning its 3'-UTR. **(I)** cDNA from HeLa stable lines expressing wt or Y306F mutant RtcB-Flag was analyzed with qPCR for the splicing of the three intron-containing tRNA molecules in human Tyr, Arg, and Ile. The graph shows the percentage of each spliced tRNA molecule to their total levels (unspliced + spliced). The presented values have been normalized to the levels of % tRNA splicing in parental HeLa cells. **(J)** The levels of the intronless tRNA molecules Pro and Val were measured by RT-qPCR in cDNA samples from HeLa stable lines expressing wt or Y306F mutant RtcB-Flag. Values have been normalized to the levels measured in Parental HeLa cells. Data information: Data values for the graphs in (A, B, D, E, I) are the mean ± SEM of three independent experiments. In (B, E) each mutant was compared with the wt using ordinary one-way ANOVA (*P < 0.05, **P < 0.01). **(G, H, J)** Data values for the graphs in (G, H, J) are the mean ± SEM of, respectively, five and four independent biological experiments. The unpaired t test was applied for the statistical analyses in those panels (*P < 0.05, ****P < 0.0001).
Source data are available online for this figure.

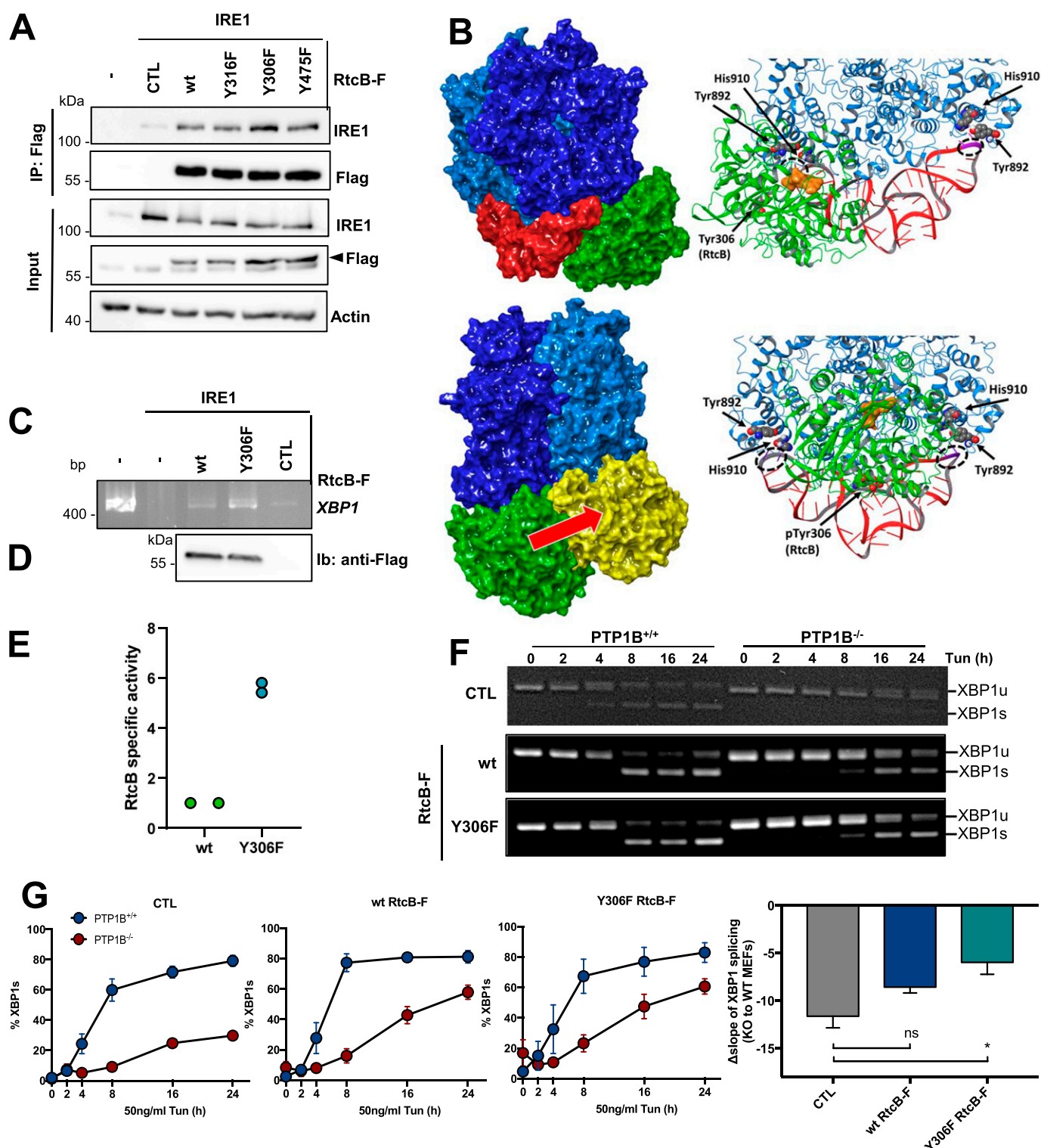

**Figure 6. Y306F RtcB mutant rescues a defective XBP1 mRNA splicing.**
**(A)** HEK cells were co-transfected with 1 μg of wt IRE1α and 1 μg of WT. or mutant Flag-RtcB plasmid. 24 h later, cell lysis and anti-Flag immunoprecipitation were performed. The immunoprecipitates were blotted for IRE1α, and the input samples for IRE1α, Flag, and actin. The arrowhead denotes the Flag-RtcB protein. **(B)** Docked complexes of RtcB WT and pY306 variants toward the IRE1 tetramer/XBP1 complex. (Top panels) RtcB WT (green) binds at the loop-binding RNase area of IRE1 (blue) and is perfectly oriented to initiate the ligation of XBP1 (red) upon its cleavage by IRE1. (Lower panels) The interaction area of the phosphorylated RtcB is shifted toward the middle of XBP1 at the IRE1 dimer–dimer interface region and will not be able to initiate the ligation reactions of the spliced ends at the two XBP1 loops (pY306 of RtcB is represented in yellow). The red arrow indicates RtcB position shift on the IRE1 tetramer when Y306 is phosphorylated. **(C)** In vitro reconstitution of XBP1 mRNA splicing using in vitro transcribed XBP1 mRNA which was incubated with or without 250 ng of the recombinant IRE1 cytosolic domain and with WT or Y306F RtcB-Flag

## Non-phosphorylable RtcB Y306 (Y306F) compensates defective XBP1 mRNA splicing and confers resistance to ER stress–induced cell death

Based on the observations obtained from (i) MD simulations that pY306 is almost totally exposed to the solvent and not buried in the protein in its fully pY status, and (ii) from the XBP1 splicing activity time-course of the respective HeLa line that Y306F RtcB-Flag contributes to more efficient XBP1 mRNA splicing, we next tested if the phosphorylation of Y306 impacted on the interaction of RtcB with IRE1. This interaction was previously shown to ensure proper XBP1 mRNA splicing (Lu et al, 2014). HEK293T cells were co-transfected with IRE1-WT and either WT or mutant forms of RtcB-Flag expression plasmids. 24 h post-transfection, cell lysates were immunoprecipitated with anti-Flag antibodies and immunoblotted with anti-IRE1 antibodies. We thus confirmed the interaction between IRE1 and RtcB (Fig 6A) and found that the Y306F RtcB-Flag mutant exhibited a stabilized interaction with IRE1 compared with the WT form (Fig 6A). The interaction of RtcB-WT and its pY306 variant with IRE1 and XBP1 mRNA were then modeled (Fig 6B). In the absence of XBP1, protein docking showed that RtcB-WT was able to bind essentially anywhere on the IRE1 tetramer surface with no specific referred site of interaction. However, upon XBP1 mRNA binding, RtcB-WT docking unveiled two symmetrically positioned sites of interaction, one at each dimer, close to the corresponding XBP1 mRNA splicing site (Fig 6B, top panels). RtcB is oriented such that Y306 does not interact explicitly with IRE1/XBP1, whereas the catalytic area of RtcB is facing directly toward the cleavage site. Upon phosphorylation of Y306, RtcB instead binds in a position between the two cleavage loops of XBP1, at the IRE1 dimer–dimer interface (Fig 6B, lower panels). The phosphate group of pY306 is pointing toward/interacting with XBP1 mRNA, and the catalytic region of RtcB is facing directly toward the IRE1 tetramer with no direct interaction toward the XBP1 mRNA. The free energies of binding as computed using the MM-GBSA theory are similar for both WT and pY306 RtcB, albeit slightly stronger for the non-phosphorylated variant (–50.2 and –45.7 kcal mol$^{-1}$, respectively). These docking calculations display a clear difference in the interaction of RtcB versus pY306 RtcB, with the WT system being perfectly oriented so as to interact with the spliced ends of XBP1 mRNA upon cleavage. Because our observations suggested that the RtcB Y306-dependent interaction could control the XBP1 mRNA splicing activity of the IRE1/RtcB complex, we thus reconstituted in vitro the splicing reaction using in vitro transcribed XBP1 mRNA, recombinant IRE1 cytosolic domain whose optimal concentration was empirically determined (Fig S7F) and RtcB immunoprecipitated

from cells. This reconstitution assay showed that as anticipated from our previous results, IRE1/RtcB Y306F–spliced XBP1 mRNA more efficiently than RtcB-WT (Fig 6C–E). This result urged us to test whether the Y306F mutant would rescue the defective XBP1 splicing observed in PTP1B$^{-/-}$ MEFs (Figs 1F and S8A and B). PTP1B$^{-/-}$ and PTP1B$^{+/+}$ MEFs were either left untransfected or transfected with RtcB-WT-Flag or Y306F RtcB-Flag plasmids (Fig S8D) and monitored for XBP1 mRNA splicing upon 0, 2, 4, 8, 16, and 24 h of 50 ng/ml Tun treatment (Fig 6F). Both WT and Y306F RtcB-Flag rescued XBP1 mRNA splicing in PTP1B$^{-/-}$ MEFs. Importantly, the quantification of these results revealed a better rescue of the Y306F RtcB-Flag, consistent with the effect on the XBP1s activity observed in stable HeLa cell lines (Fig 6F and G). The rescuing effect of Y306F RtcB-Flag was profound even under acute ER stress induced by 10 µg/ml Tun for 6 h (Fig S8C). The importance of the phosphorylation event on the Y306 site was also reinforced by the docking of c-ABL on RtcB using MD simulation (Fig S8E). The interacting residues on c-ABL (D325, D444) and RtcB (K279, R283, K357) created a surface of contact where Y306 of RtcB is well accessible to the active site of the kinase (Fig S8F).

On the grounds of the pro-survival character of *XBP1* mRNA splicing, we then wondered whether the phosphorylation of Y306 in RtcB could have an effect on the cellular survival/death. When cells stably over-expressing the WT or the Y306F form of RtcB were treated for 24 h with increasing concentrations of DTT to induce ER stress, the survival advantage of the latter cells became apparent (Fig 7A). Y306F RtcB-Flag–expressing cells presented even less necrosis when endogenous RtcB was knocked down by using an siRNA targeting the 3'-UTR of the RtcB mRNA. This meant that the observed effect was solely due to the exogenously expressed Y306F RtcB mutant (Fig 7A). Under these specific stress conditions, in contrast to necrosis/late apoptosis, early cell apoptosis was not evident. Efficient silencing of endogenous RtcB was confirmed by both qPCR using primers spanning the 3'-UTR of the RtcB mRNA (Fig S8G) and Western blot analysis (Fig S8H). These results indicate that RtcB Y306 is a key residue in the IRE1-XBP1/RIDD signaling because it is phosphorylated by c-ABL and dephosphorylated by PTP1B, and it regulates the interaction between RtcB and IRE1. Further to this, we realize that the phosphorylation of RtcB on Y306 is able to direct a decision toward cell death in the presence of ER stress downstream of the IRE1 RNase signaling outputs. To further document the role of the pY-dependent interaction between RtcB and IRE1 in the control of cell death, we relied on the recent discovery that DNA-damaging agents signaled through RIDD (Dufey et al, 2020) and hypothesized that forcing the splicing of XBP1 mRNA through the expression of Y306F RtcB should sensitize the cells to cell death. To test this, cells expressing wt or Y306F RtcB

---

immunoprecipitated from cells. The retrotranscribed XBP1 DNA from the assay was analyzed by PCR using primers recognizing the XBP1 mRNA. **(D)** Immunoprecipitated RtcB levels were detected by immunoprecipitating the cell lysates using anti-Flag antibody–conjugated beads and immunoblotting of the immunoprecipitates with anti-Flag antibodies. **(C, D, E)** XBP1s (C) was then normalized to RtcB protein levels (D) to obtain the RtcB-specific activity. **(F)** PTP1B$^{+/+}$ (WT) and PTP1B$^{-/-}$ (KO) MEFs untransfected (CTL) or transfected with 2 µg of Flag-RtcB-WT or Flag-RtcB-Y306F plasmid were treated 24 h post-transfection with 50 ng/ml Tunicamycin for 0, 2, 4, 8, 16, and 24 h. Their cDNA was analyzed by PCR using primers recognizing the *XBP1* mRNA. **(F, G)** Quantification of gels in (F). The graph shows the comparison of XBP1 mRNA splicing in CTL cells and cells expressing either Flag-RtcB-WT or Flag-RtcB-Y306F. The composite comparison of the results obtained for ctl (circles) or rescues with either wt RtcB (triangles) or Y306F RtcB (squares) is also shown. (Bar graph) The slope of each curve was calculated, and for each condition, the slope of the WT cells was subtracted from the one of the KO cells, called as Δslope. The calculation of the Δslope was also corrected by the expression levels of Flag-RtcB-WT or Flag-RtcB-Y306F as determined by Western blotting. Data information: Data values presented in (E) represent two independent experiments. Data values in (G) are the mean ± SEM of three independent experiments. The unpaired *t* test was applied for the statistical analyses (ns, nonsignificant, *P < 0.05, **P < 0.01).
Source data are available online for this figure.

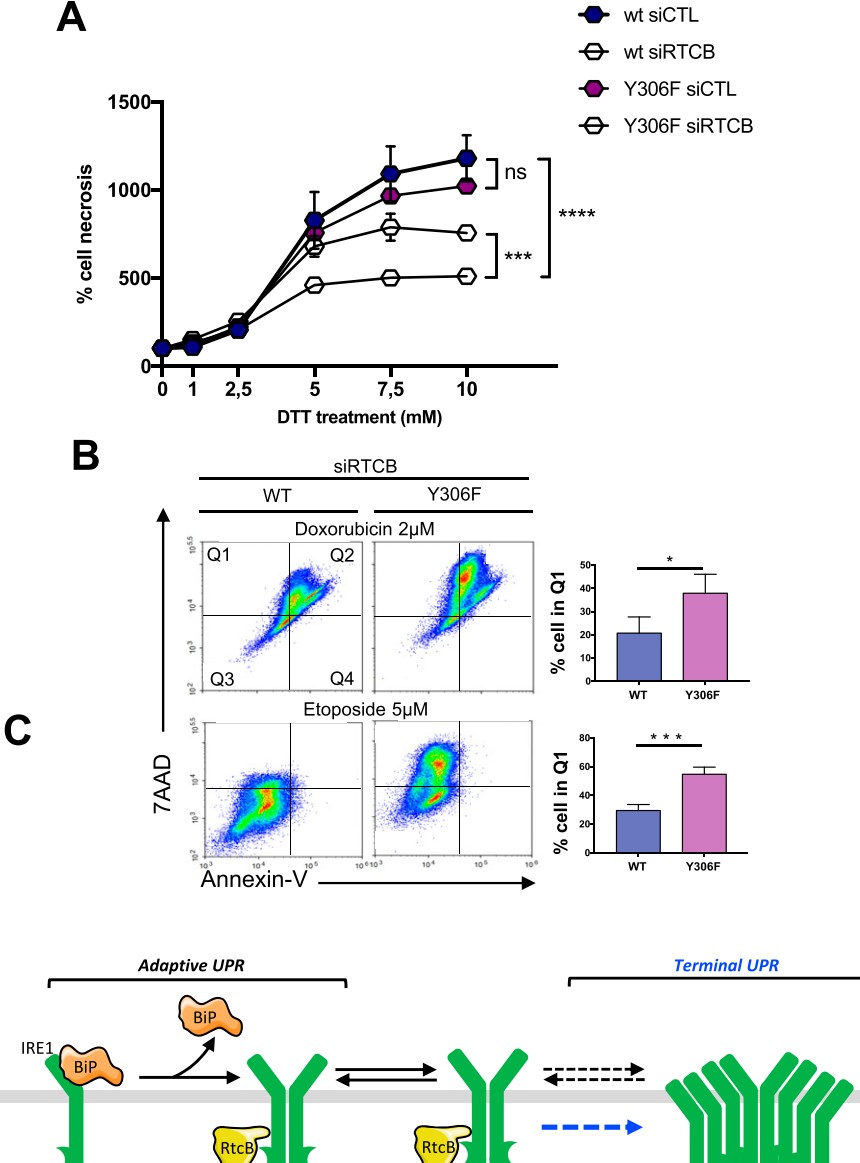

**Figure 7. RtcB Y306 phosphorylation is a key event in modulating stress-induced cell life and death decisions.**
**(A)** HeLa lines stably expressing wt or Y306F RtcB-Flag were transfected or not with siRNA sequences against the endogenous RTCB mRNA. 48 h post-transfection, they were treated with 0, 1, 2.5, 5, 7.5, and 10 mM DTT for 24 h. The resulting samples were then analyzed by FACS for cell necrosis and apoptosis through 7-AAD and Annexin V staining, respectively. Data values are the mean ± SEM of n = 4 independent experiments. Two-way ANOVA and Tukey's multiple comparisons test was applied for the statistical analyses (ns, nonsignificant, ***$P$ < 0.001, ****$P$ < 0.0001). **(B)** HeLa lines stably expressing Flag-RtcB-WT or Flag-RtcB-Y306F were left untransfected or transfected with siRNA sequences against the endogenous RTCB mRNA. 48 h post-transfection, they were treated with 0, 0.5, 1, and 2 $\mu$M doxorubicin, or 0, 5, 10, and 20 $\mu$M etoposide for 24 h. The resulting samples were then analyzed by FACS for cell necrosis and apoptosis through 7-AAD and Annexin V staining, respectively. Data from 2 $\mu$M doxorubicin and 5 $\mu$M etoposide treatments are shown. Data values are the mean ± SEM of n = 4 independent experiments in two-way ANOVA and post hoc Tukey multiple comparisons test (*$P$ < 0.05, ***$P$ < 0.001). **(C)** Model representation of IRE1 activation and signaling toward XBP1s or regulated IRE1-dependent decay (RIDD). A decrease in PTP1B expression driven by RIDD is thought to reduce the formation of the IRE1/RtcB complex, thereby pushing toward unleashed RIDD and terminal unfolded protein response. Dashed lines represent a higher order oligomerization of IRE1, which might result in terminal unfolded protein response. Source data are available online for this figure.

were treated with doxorubicin or etoposide, and in both cases, Y306F RtcB–expressing cells were more prone to cell death than cells expressing wt RtcB (Figs 7B, S9, and S10), thereby indicating that the balance between XBP1 mRNA splicing and RIDD (i) is tightly controlled by RtcB, (ii) plays a key role in stress-dependent life and death decisions, and (iii) determines the nature of the biological output depending on the stress cells are exposed to.

## Discussion

The objective of this study was to better characterize how *XBP1* mRNA splicing and RIDD activities can cross talk and how their signals are integrated. We showed that RtcB, the tRNA ligase responsible for XBP1 mRNA splicing, is at the center of a kinase/phosphatase network and that the tyrosine phosphorylation–

dependent regulation of RtcB interaction with IRE1 might represent a mechanism to shift IRE1 activity from XBP1 mRNA splicing to RIDD. We have identified c-ABL as a tyrosine kinase causing RtcB phosphorylation on the three sites: Y306, Y316, and Y475, and PTP1B as a tyrosine phosphatase de-phosphorylating RtcB. Interestingly, the phosphorylation of Y475 on RtcB, a residue located in the active site, attenuated XBP1 mRNA splicing. Indeed, the phosphorylation of Y475 prevents GTP binding to His 428, thus halting the process of ligation. More surprisingly, the phosphorylation of Y306 weakened the interaction between RtcB and IRE1, thus reducing XBP1 mRNA splicing activity. This could in turn favor the formation of IRE1 oligomers, leading to the observed increase in RIDD activity. In the same line of evidence, the expression of RtcB-Y306F compensated defective XBP1 mRNA splicing observed in PTP1B$^{-/-}$ cells, thereby indicating that the phosphorylation on this specific residue impacts negatively on the XBP1 mRNA splicing activity and constitutes a PTP1B-dependent mechanism in the UPR. Moreover, the fact that RtcB-Y306F conferred resistance to ER stress–induced cell death poses Y306 phosphorylation (preceding or following the phosphorylation on Y316 and Y475) as a key cellular event, allowing life/death decisions (Fig 7C).

Our data confirm our initial hypothesis by demonstrating the existence of an autoregulatory network emanating from the IRE1 RNase activity. In this regulatory network, PTP1B was identified as a key actor by dephosphorylating RtcB, thereby facilitating the activation of the IRE1/XBP1s axis. PTP1B mRNA is a RIDD substrate such that when RIDD becomes dominant, PTP1B expression is prevented. Ultimately, one could propose a new model in which PTP1B would keep RtcB in a non-phosphorylated state, thereby allowing the IRE1/RtcB complex to efficiently splice XBP1 mRNA, hence exerting adaptive UPR. However, when the stress cannot be resolved, PTP1B expression would decrease through a RIDD-dependent mechanism. In parallel, c-ABL is recruited to the ER membrane (Morita et al, 2017) where it phosphorylates RtcB on Y306, Y316, and Y475, thus further inhibiting IRE1/RtcB interaction initially lowered by the reduction of PTP1B expression and, in the meantime, enhancing RIDD by prompting IRE1 oligomerization, hence resulting in terminal UPR. Moreover, the relationship between IRE1 activity triggered by ER stress and the IRE1 oligomeric state that was recently reported (Belyy et al, 2021 *Preprint*) is in accordance with our model in which XBP1 mRNA splicing might be catalyzed by tetrameric oligomers. An integrated model could propose that ER stress–driven tyrosine phosphorylation of RtcB might lead to its dissociation from the IRE1 splicing complex, which in turn would allow the formation of active IRE1 oligomers, thus activating RIDD.

In line with these observations, it was demonstrated in a recent report that c-ABL can dissociate from 14-3-3 proteins in the cytosol upon ER stress and be recruited to IRE1 foci at the ER membrane augmenting terminal UPR (Morita et al, 2017). Herein, we show that c-ABL can act not only as a scaffold for IRE1 but also through its catalytic activity with the phosphorylation of RtcB on tyrosine residues that in turn causes the reduction in its *XBP1* mRNA splicing activity. From our results and other studies, we can also speculate that apart from c-ABL, other tyrosine kinases might be involved in the regulation of the UPR. For instance, the SRC tyrosine kinase has

recently been reported to be activated by and bind to IRE1 under ER stress, leading to ER chaperone relocalization to the cell surface (Tsai et al, 2018). As such, the scaffolding role of IRE1 could represent a means to integrate multiple signaling pathways thus far unrelated to IRE1 signaling and link them to a clear function in cell homeostasis.

In the present study, we identified RtcB as a new target of the tyrosine phosphatase PTP1B. Interestingly, besides our initial report that PTP1B is involved in the control of IRE1 activity (Gu et al, 2004), PTP1B was subsequently found to also target the PERK arm of the UPR by negatively regulating its activation through direct dephosphorylation (Krishnan et al, 2011). As such, one might consider tyrosine kinase/tyrosine phosphatase signaling as a new key element in the regulation of the UPR to control life and death decisions. Our findings indicate that the exchange of a phosphatase for a kinase, and vice versa, is crucial for the XBP1 mRNA splicing catalyzed by IRE1 and RtcB. In the molecular complex of IRE1 and RtcB, an additional member could potentially be the archease protein reported as a stimulatory co-factor of RtcB not only for tRNA but also for *XBP1s* mRNA ligation (Jurkin et al, 2014; Popow et al, 2014). Our results demonstrate that the dynamic alteration of the constituents of a multi-protein complex at the ER membrane can drive its collective activities, which further supports the notion of a UPRosome in the control of cellular life and death decisions (Hetz & Glimcher, 2009; Hetz & Papa, 2018).

# Materials and Methods

## Materials

Tun, DTT, actinomycin D, cycloheximide, and sodium orthovanadate (Na$_3$VO$_4$) were obtained from Sigma-Aldrich. The phosphatase inhibitor bpV(phen) and the tysosine kinase inhibitors afatinib, crizotinib, and sorafenib were from Santa Cruz Biotechnology. The tyrosine kinase inhibitors dasatinib and imatinib (imatinib mesylate; STI571) were from Selleckchem. Protease and phosphatase inhibitor cocktail tablets were from Roche through Sigma-Aldrich. Pierce ECL Western Blotting Substrate was from Thermo Fisher Scientific. The pCMV3-mRTCB-Flag plasmid encoding the wt form of mouse RtcB was purchased from Sino Biological. Various mutant forms of this plasmid were made with the QuikChange II XL Site-Directed Mutagenesis Kit from Agilent using the appropriate primers each time (Table S4). The pcDNA3.1/Zeo-PTP1B wt and pcDNA3.1/Zeo-PTP1B C215S plasmids were used as previously described (Blanchetot et al, 2005). The pCDH-CMV-IRE1α-EF1-Puro-copGFP was cloned in the laboratory, as previously described (Lhomond et al, 2018).

## Cell culture and transfection

HEK293T, HeLa cells, U87 cells bearing a dominant negative form of IRE1α (IRE1 DN) or an empty vector (Drogat et al, 2007) and MEFs wt or KO for PTP1B (obtained in Gu et al [2004]) were cultured in DMEM supplemented with 10% FBS at 37°C in a 5% CO$_2$ incubator. For the

generation of HeLa cell lines stably expressing each one of the RtcB-Flag proteins, cells were first transfected using Lipofectamine 2000 (Thermo Fisher Scientific) with 1 $\mu$g of the corresponding plasmid. After 24 h, the medium of the cells was changed to medium containing 600 $\mu$g/ml Hygromycin B (Thermo Fisher Scientific) to start the selection process. The Hygromycin B–containing medium was replaced every 3 d for a total period of 21 d, when polyclonal populations of HeLa cells stably expressing each of the RtcB-Flag proteins were obtained. The stable cell lines were maintained in DMEM with 10% FBS containing 120 $\mu$g/ml Hygromycin B. Transient transfections were achieved using either polyethylenimine (PEI) for the HEK293T cells or Lipofectamine 2000 for the HeLa cells and the MEFs together with the desired plasmid. For preparation of the PEI solution, branched polyethylenimine (average $M_w$~25,000) powder was purchased from Sigma-Aldrich.

### siRNA-based screening assay

The library of ~300 siRNAs against genes encoding ER proteins was used in previous studies (Higa et al, 2014). Each siRNA (25 nM) was transfected into HEK293T cells by using the Lipofectamine RNAiMAX reagent (Thermo Fisher Scientific). The cells were then transfected with an XBP1s-luciferase reporter in which the firefly luciferase gene is fused to the XBP1 gene (Spiotto et al, 2010): under basal conditions where XBP1 mRNA remains unspliced, a stop codon at the gene fusion is in-frame, thus not allowing the luciferase to be expressed. In contrast, under ER stress conditions, the splicing of XBP1 mRNA shifts the open reading frame (ORF), resulting in the loss of the stop codon downstream of XBP1 and the subsequent expression of luciferase. ER stress was then applied to the cells and sequentially the substrate of the luciferase enzyme, luciferin. The cells were then tested for the emission of light that after appropriate quantification and a value threshold setup led to the distinction of two major groups of genes: XBP1s-positive and XBP1s-negative regulators. High values of luminescence (>1 AU) signify that efficient XBP1 splicing took place, so the luciferase expression was allowed, thus marking the silenced genes as negative XBP1s regulators. The opposite is true for values <1 AU that place the silenced genes in the category of positive regulators of XBP1 mRNA splicing.

### In vitro IRE1 mRNA cleavage assay

As described previously (Lhomond et al, 2018), in this assay, total RNA was extracted from U87 cells, refolded and incubated or not with recombinant IRE1$\alpha$ protein under appropriate physicochemical conditions. The resultant RNA sequences from the cleaved and uncleaved conditions underwent polyA mRNA isolation, the final different groups (non-polyA: cleaved or uncleaved, and polyA: cleaved or uncleaved) were reverse transcribed, and the respective cDNA material was hybridized on a human transcriptome array whose analysis revealed potential RIDD substrates.

### Immunoblot and immunoprecipitation (IP)

For preparation of whole cell extracts, cells were incubated in RIPA buffer (50 mM Tris–HCl pH 7.5, 150 mM NaCl, 1% NP-40, 0.5% sodium deoxycholate, 0.1% SDS, 2 mM EDTA, and 50 mM NaF) supplemented with protease and phosphatase inhibitors, for 20 min on ice. Cells were scraped and centrifuged at 4°C for 7 min at 17,000$g$ to collect the supernatant containing the protein. The samples were applied to SDS–PAGE and analyzed by immunoblotting. Dilutions of primary antibodies used for immunoblotting were as follows: mouse monoclonal anti-FLAG M2 (Sigma-Aldrich), 1:2,000; rabbit polyclonal anti-RtcB (ProteinTech), 1:1,000; for recognition of the human PTP1B, mouse monoclonal anti-PTP1B (3A7) (Santa Cruz Biotechnology), 1:1,000; for recognition of the mouse PTP1B, rabbit polyclonal anti-PTP1B (ab88481) (Abcam), 1:1,000; rabbit polyclonal anti-IRE1$\alpha$ (B-12) (Santa Cruz Biotechnology), 1:1,000; mouse cocktail anti-pY-HRP (PY-7E1, PY20) (Invitrogen/Thermo Fisher Scientific), 1:2,000; goat polyclonal anti-Actin (I-19) (Santa Cruz Biotechnology), 1:1,000; mouse monoclonal anti-VCP (BD Transduction Laboratories), 1:1,000; and rabbit polyclonal anti-calnexin (kindly provided by Dr. John Bergeron, McGill University), 1:1,000. Polyclonal goat anti-mouse, goat anti-rabbit, and rabbit anti-goat secondary antibodies conjugated to HRP (Dako-Agilent) were used at a dilution of 1:7,000, 1:7,000, and 1:3,500, respectively. For the IP, the cells were lysed in CHAPS buffer (30 mM Tris–HCl pH 7.5, 150 mM NaCl, and 1.5% CHAPS) supplemented with a cocktail of protease and phosphatase inhibitors. The cell lysis buffer was further supplemented with 1 mM of the phosphatase inhibitor $Na_3VO_4$ when the immunoprecipitates were tested for pTyr. The lysates were then incubated for 16 h at 4°C with the indicated IP antibody (1 $\mu$g Ab/1,000 $\mu$g protein). After this, dynabeads protein G (Thermo Fisher Scientific) were first washed with CHAPS lysis buffer and/or PBS, then mixed with the protein/Ab mixture, incubated at room temperature for 20 min or at 4°C for 40 min with gentle rotation, and washed with CHAPS and/or PBS. Finally, the beads were eluted with 1× Laemmli sample buffer (LSB), heated at 55–95°C for 5 min, and loaded to SDS–PAGE. For the immunoblotting, the suitable primary and secondary antibodies were used.

### Generation of pY475-RtcB–specific antibodies

These antibodies were generated by Biotem (France). In brief, two rabbits were immunized with a KLH-conjugated MEEAPES(Phospho)YKNVTDVV peptide (four immunizations). Sera were collected 42 d after immunization and specific antibodies subjected to affinity purification using MEEAPES(Phospho)YKNVTDVV-conjugated beads as an affinity matrix. The bound material was eluted using acidic pH and then counter-depleted for antibodies against the non-phosphorylated peptide using MEEAPESYKNVTDVV-conjugated beads. The flow-through was collected and further validated (Fig S3).

### RT-PCR and RT-qPCR

Total RNA was extracted from cells using TRIzol (Thermo Fisher Scientific) according to the manufacturer's instructions. cDNA was synthesized from the total RNA using the Maxima Reverse Transcriptase enzyme, random hexamer primers, dNTP mix, and the Ribolock RNase inhibitor (Thermo Fisher Scientific). PCR was performed on the template cDNA using Phusion High-Fidelity DNA Polymerase and dNTP mix (Thermo Fisher Scientific). Quantitative PCR was alternatively performed for the cDNA using the SYBR Premix Ex Taq (Tli RNase H Plus) (TAKARA-Clontech) using a QuantStudio5 system (Applied Biosystems). The primer sequences used for these experiments were synthesized by Eurogentec and are shown in Table S4.

## Cell death assay

HeLa cells stably expressing RtcB-Flag proteins were left untransfected or transfected with 10 nM of a duplex siRNA sequence (Table S4; IDT, Integrated DNA Technologies) targeting the 3′-UTR of the RTCB mRNA. 48 h later, the cells were treated with 0, 1, 2.5, 5, 7.5, and 10 mM DTT or 0, 0.5, 1, and 2 $\mu$M doxorubicin or 0, 5, 10, and 20 $\mu$M etoposide for 24 h. The cells were then collected in tubes, along with the corresponding supernatants and PBS washes, and centrifuged at 1,700 rpm or 750$g$ for 5 min. The cell pellets were transferred in a round-bottom 96-well plate, where 1× Annexin V buffer and Annexin V were added. After a 15-min incubation at room temperature and a PBS wash, 2% FCS in PBS and 7-AAD were added to the samples. The latter was incubated at room temperature for five more minutes and then analyzed by FACS (NovoCyte 3000).

## In vitro kinase assays

To perform an in vitro kinase assay, we used 0.36–0.5 $\mu$g of human recombinant His-tagged ABL (His-ABL1 or c-ABL: 126 kD) from Carna Biosciences as the tyrosine kinase and 1 $\mu$g of human recombinant GST-tagged RtcB (GST-C22orf28: 81.6 kD) (P01) from Abnova as the substrate. These were incubated in 1× kinase buffer (25 mM Tris–HCl pH 7.5, 10 mM MgCl$_2$, 1 mM MnCl$_2$, 0.5 mM DTT, 10 $\mu$M ATP, 0.1 mM Na$_3$VO$_4$, and 5 mM $\beta$-glycerophosphate) in a total volume of 30 $\mu$l at 37°C for 30 min or at RT for 2 h. Upon completion of the reaction, 5× LSB was added to the reaction samples which were heated to 100°C and applied in 6% SDS–PAGE.

## In vitro reconstitution of XBP1u splicing

To reconstitute XBP1u splicing in vitro, a vector encoding XBP1u downstream of T7 promoter was used as template for the splicing substrate (plasmids were a kind gift from Dr Fabio Martinon, Lausanne, Switzerland). The splicing substrate XBP1u mRNA was prepared using a RiboMAX Large Scale RNA Production Systems—T7 kit (Promega) at 37°C for 3 h. In vitro transcripts were purified using an RNeasy kit (QIAGEN). Flag-RtcB WT and Flag-RtcB Y306F proteins were purified from HeLa cells stably expressing the two proteins, respectively. Cell pellets were lysed in a buffer containing 30 mM Tris–HCl, pH 7.5, 150 mM NaCl, and 1.5% CHAPS supplemented with a cocktail of protease and phosphatase. Flag-tagged RtcB proteins were affinity-purified from the lysates with anti-Flag M2 affinity gel (Sigma-Aldrich), followed by extensive washes. The human recombinant IRE1 cytoplasmic domain was purchased from Sino Biological. The reconstituted in vitro XBP1u splicing assay was carried out at 37°C for 2 h in kinase buffer (2 mM ATP, 2 mM GTP, 50 mM Tris–HCl pH 7.4, 150 mM NaCl, 1 mM MgCl$_2$, 1 mM MnCl$_2$, 5 mM $\beta$-mercaptoethanol). 1 $\mu$g XBP1u RNA, 250 ng IRE1 protein, and IP-purified Flag-RtcB on beads were used for a 50 $\mu$l splicing reaction. The spliced products were column-purified using an RNeasy kit (QIAGEN) for RT-PCR analysis. The PCR products were resolved on 3% agarose gel.

## Mass spectrometry analyses

Samples for the mass spectrometry analysis were obtained from a scaled up in vitro kinase reaction containing 0.9 $\mu$g of human recombinant His-c-ABL (His-ABL1 or c-ABL: 126 kD) and 5 $\mu$g of human recombinant GST-RtcB in the same conditions as described

in the previous section. The difference is that the samples were ran on a precast 8% polyacrylamide gel (Eurogentec), which was afterward stained with 10% CBB and destained to be sent for mass spectrometry analysis. At the proteomics platform, the gel was re-stained with Page Blue Protein Staining Solution (Thermo Fisher Scientific). In-gel digestion was then performed with trypsin (cleavage site = K/R) or lysC (cleavage site = K) and TiO$_2$ purification (10% not purified as control) followed. For the mass spectrometry (MS) analysis, the recovery volume was 15 $\mu$l 0.1% TFA and the injection volume was 5 $\mu$l. The instrument linear ion trap (LTQ) Velos and the PepMap 25-cm or Orbitrap ELITE/C18 Accucore 50-cm columns were used. 1 h run mass spectrum analyzer and higher-energy C-trap dissociation (HCD) method were used. For the data processing, software Proteome Discoverer 2.1 (Sequest HT/Percolator) was used, with the thresholds of 1% false discovery rate, ptmRS for site assignation. Data are available via ProteomeXchange with identifier PXD023433.

## Modeling of WT and phosphorylated RtcB

The starting structure for the MD simulations was a homology model of the human RtcB protein (Nandy et al, 2017). For the modified versions of the protein regarding its tyrosine phosphorylation states, we edited the initial structure through YASARA (Yet Another Scientific Artificial Reality Application, v15.3.8) (Krieger & Vriend, 2014). For the MD simulations using Gromacs 5.1 (Abraham et al, 2015), preparation and generation of additional missing parameters to the AMBER ff14SB force field (Maier et al, 2015) were performed for each structure using the AmberTools17 package (Case et al, 2017). Because RtcB contains a Mn$^{2+}$ ion in its active site, we employed the Python-based metal center parameter builder MCPB.py (version 3.0) (Li & Merz, 2016) included in AmberTools17 for the parameterization of the metal site. In addition, separate pdb files were generated for the nonstandard amino acid of phosphorylated tyrosine (named "TYP") and GMP bound to H428 (named "HIG"), using standard protocols. The resulting topology and coordinate files from Amber were transformed to the respective Gromacs files to perform the necessary system preparations for the MD simulations. Each system was solvated with TIP3P water (Jorgensen et al, 1998) under periodic boundary conditions with the minimum distance between any atom in the solute and the edge of the periodic box being 10.0 Å. Na$^+$/Cl$^-$ counterions were added as appropriate for the neutralization of the system. Energy minimization was conducted until the force was <1,000 kJ mol$^{-1}$ nm$^{-1}$, followed by a 100-ps NVT and a 100-ps NPT equilibration, and a final 200-ns classical MD simulation for each system, using the Gromacs 5.1 package (Abraham et al, 2015). The temperature was kept at 300 K by the velocity rescaling thermostat (Bussi et al, 2009), with a coupling constant of 0.1 ps. In the NPT equilibration and MD simulation, the pressure was kept at 1.0 bar using the Parrinello–Rahman barostat (Parrinello & Rahman, 1981) with a coupling time of 2.0 ps. The leap-frog algorithm (Van Gunsteren & Berendsen, 1988) was used with an integration time-step of 2 fs, and the LINCS algorithm (Hess et al, 1997) was used to apply constraints on all bonds. UCSF Chimera (Pettersen et al, 2004) was used to visualize the processed systems. For quality control of our simulations, we calculated the RMSF and RMSD of our systems. Post-MD analyses also included calculation of the surface solvent accessible area via

Gromacs, pKa values, and percentage of surface/buried area of the key tyrosine residues via the PROPKA 3.0 part of the PDB2PQR web server 2.0.0 (Dolinsky et al, 2004). To determine the relative positions of the tyrosines of interest and illustrate if and how much they are rotated, we set as a fixed point the $Mn^{2+}$ ion in the active site of RtcB and calculated the dihedral angles consisting of this metal ion and the CA, CB, and O atoms of the tyrosine residue using UCSF Chimera. For the representation of the dihedral angles, polar charts were created using the matlablib 2.1.2 in Python (Droettboom et al, 2018). Furthermore, we calculated the docking score of the ligand GTP in the active site of the different RtcB systems (without GMP in the active site). For this purpose, among the programs of the Schrödinger Suites 2018-1 package, we used Protein Preparation Wizard (Madhavi Sastry et al, 2013) to prepare the protein structure, LigPrep to prepare the GTP ligand, and Glide (Friesner et al, 2006) for receptor grid generation and XP ligand docking. The grid in which the ligand was allowed to dock was determined by analyzing the interactions between GMP and its surrounding amino acids in MOE (Molecular Operating Environment, 2016.01; Chemical Computing Group) (Molecular Operating Environment (MOE), 2016).

**Modeling of protein complexes**

To explore the interaction of RtcB with the IRE1-XBP1 complex, we used a recent model of the human IRE1 tetramer (dimer of dimers) (Carlesso et al, 2020), and the homology model of human RtcB (Nandy et al, 2017). The RNAComposer modeling webserver (http://rnacomposer.ibch.poznan.pl) (Popenda et al, 2012) was used to predict the 3D structure of XBP1 mRNA based on the nucleotide sequence and its secondary structure (Peschek et al, 2015). The predicted model was refined and optimized using 3dRNA v.2.0 (http://biophy.hust.edu.cn/3dRNA) (Wang et al, 2019). The HAD-DOCK (Van Zundert et al, 2016) webserver was employed to predict the IRE1-tetramer/XBP1 complex. The cleavage site in both stem loops of XBP1 mRNA (G and C nucleotides) (Popenda et al, 2012) and active residues in the RNase domain of the IRE1 tetramer (i.e., His910 and Tyr892) (Sanches et al, 2014) were defined as interacting regions in HADDOCK. For the docking of RtcB and its pTyr306 phosphorylated variant to either the IRE1 tetramer or the IRE1 tetramer/XBP1 complex, the webserver HDOCK (Yan et al, 2017) was employed, using a blind docking strategy. The Schrödinger package (release 2020-4; Schrödinger Inc) was employed to calculate the free energy of binding in IRE1-tetramer/XBP1 and IRE1-XBP1/RtcB complexes using the molecular mechanics generalized Born surface area technique (Rastelli et al, 2010). For the c-Abl–RtcB interaction, we used the active form of Abl (PDB-ID: 2GQG). The ATP molecule was aligned inside protein using Chimera, based on the crystal structure of ATP bound to the structurally similar kinase Src (PDB-ID: 5XP7). The proteins were docked using PatchDock (Schneidman-Duhovny et al, 2005), with Y306 (RtcB) and ATP (c-ABL) chosen as interacting units. Following visual inspection of obtained complexes, the best aligned complex was prepared for MD simulations as outlined before. For the complex to not immediately dissociate, a set of seven position-restrained equilibrations, with gradually reduced restraints in each, were applied. Finally, the system was subjected to a 200-ns MD simulation without position restraints.

**Bioinformatics tools**

For the creation of Venn diagrams, we used the online tool http://bioinformatics.psb.ugent.be/webtools/Venn/. For the creation of gene networks and functional annotations, we used the online String database 11.0 (https://string-db.org/) (Jensen et al, 2009). Multiple sequence alignments were performed using Clustal Omega 1.2.4 (Sievers et al, 2011). The PhosphoSitePlus database was used for the search of reported tyrosine phosphorylation sites on the proteins of interest (Hornbeck et al, 2015).

**Statistical analyses**

*T* test, one-way ANOVA, and two-way ANOVA were applied for the statistical analyses depending on the experimental setting through GraphPad Prism 7.0a software.

# Data Availability

Proteomics data are available via ProteomeXchange with identifier PXD023433.

# Supplementary Information

# Acknowledgements

We thank Dr. Anna Reymer, Dr. Antonio Carlesso, Dr. Samuel Genheden, and Johanna Hörberg for training on computational chemistry/biology (UGOT), and Dr. Aeid Igbaria for critical reading of the manuscript. This work was funded by grants from Institut National du Cancer (PLBIO), SIRIC-ILIAD/Cancéropôle Grand Ouest to E Chevet, Fondation pour la Recherche Médicale (équipe labellisée 2018) to E Chevet and R Pedeux; by European Union (EU) H2020 MSCA ITN-675448 (TRAINERS), and MSCA RISE-734749 (INSPIRED) grants to E Chevet and LA Eriksson. The Swedish Research Council (VR) and the Swedish National Infrastructure for Computing are gratefully acknowledged for funding and allocations of computing time at the C3SE and PDC supercomputing centers, respectively (LA Eriksson). A Papaioannou is a Marie Curie early-stage researcher funded by EU H2020 MSCA ITN-675448 (TRAINERS). A Metais was funded by the Fondation ARC pour la recherche contre le cancer. SJ Mahdizadeh was funded by the Vinnova Seal-of-Excellence programme 2019-02205 (CaTheDRA).

**Author Contributions**

A Papaioannou: conceptualization, formal analysis, investigation, visualization, methodology, and writing—original draft.
F Centonze: conceptualization, formal analysis, validation, investigation, visualization, methodology, and writing—review and editing.
A Metais: investigation.
M Maurel: conceptualization and investigation.
L Negroni: visualization and methodology.
M Gonzales-Quiroz: investigation, visualization, and methodology.

SJ Mahdizadeh: formal analysis, investigation, and visualization.

G Svensson: formal analysis, investigation, and visualization.

E Zare Golchesmeh: formal analysis, investigation, and visualization.

A Blondel: formal analysis, investigation, and visualization.

AC Koong: resources.

C Hetz: conceptualization, resources, and funding acquisition.

R Pedeux: conceptualization and funding acquisition.

ML Tremblay: resources.

LA Eriksson: conceptualization, data curation, formal analysis, supervision, funding acquisition, investigation, visualization, methodology, and writing—review and editing.

E Chevet: conceptualization, formal analysis, supervision, funding acquisition, validation, investigation, project administration, and writing—original draft, review, and editing.

## Conflict of Interest Statement

E Chevet and LA Eriksson are founders of Cell Stress Discoveries Ltd. E Chevet is founder of Thabor Therapeutics. The authors declare no conflicting interests.

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
