## [Reviewer comments · Life Science Alliance]

Life Science Alliance

Stress-induced tyrosine phosphorylation of RtcB modulates IRE1 activity and signaling outputs.

Alexandra Papaioannou, Federica Centonze, Alice Metais, Marion Maurel, Luc Negroni, Matias Gonzales-Quiroz, Sayyed Mahdizadeh, Gabriella Svensson, Ensieh Zare, Alice Blondel, Albert Koong, Claudio Hetz, Rémy Pedeux, Michel Tremblay, Leif Eriksson, and Eric Chevet

DOI: <https://doi.org/10.26508/lsa.202201379>

Corresponding author(s): Eric Chevet, Inserm and Leif Eriksson, Univ Gothenborg

Review Timeline:

Submission Date:	2022-01-20
Editorial Decision:	2022-01-21
Revision Received:	2022-01-23
Editorial Decision:	2022-01-25
Revision Received:	2022-01-27
Accepted:	2022-01-31

Transaction Report:

Please note that the manuscript was previously reviewed at another journal and the reports were taken into account in the decision-making process at *Life Science Alliance*.

Referee #1

Review

Report for Author:

Since the authors have revised the manuscript in accordance with my comments, Reviewer 1 thinks the manuscript is now suitable for publication in this journal.

Referee #2 Review

Report for Author:

The authors have addressed my concerns in the revised manuscript

Referee #3 Review

Report for Author:

Papaioannou and colleagues have responded to my comments but, unfortunately, have not satisfactorily resolved the issues I viewed as major (see detailed explanation below). While the proposed model remains intriguing, it is not convincingly supported by the evidence presented and I therefore cannot recommend publication in this journal.

Major points

Comment 1 (Figure 1E)

In the re-plotted graph, the difference in Xbp1 splicing between tunicamycin-treated control and PTPB1 knockout cells is very minor (seven-fold versus six-fold above untreated control cells). It is not clear if this effect is meaningful or even statistically significant. The figure legend states that "Data values are the mean {plus minus} SEM of 4 independent experiments. Unpaired t test was applied for the statistical analyses." However, it is unclear what the asterisks shown in Figure 1 E refer to, i.e. it is not

indicated which sample was tested against which by a t-test. Also, it is not stated if the t-test used assumed equal or unequal variance, which is important because the data are normalized. Finally, text states that "... this was reversed upon treatment with TM for 3 and 6 hours" but there are no data for the 3 h time point in Figure 1E. Because of these problems, it remains unclear if stress-induced Xbp1 signaling is impaired in PTP1B knockout cells. Figure S1E-G does not help to clarify the issue because the experiment shown there was done only once.

Comment 2 (Figure 5A)

The authors re-plotted the data as suggested and show them in the point-by-point response as Figure R3.1. This graph clearly shows that even at the most favorable time point (8 h of tunicamycin treatment) there is no statistically significant difference between cells expressing RtcB WT and RtcB Y306F (second and third bar). The authors do not provide any statistics but judging from the individual data points, the values for cells expressing RtcB WT are approximately 2, 3.8 and 5. The values for the cells expressing RtcB Y306F are approximately 3, 5.7 and 6.5. Testing these data sets against each other with a two-tailed t-test assuming equal variance yields $p > 0.3$. The data therefore do not allow the conclusion that cells expressing RtcB WT and RtcB Y306F are different.

Comment 3 (Figure 5C)

The new blot shown in Figure 5C still shows saturated signal in the panel with the anti-FLAG antibody (see lanes 2 and 5), so any quantification based on blots like this is problematic.

Comment 4 (Figure 5D)

To my mind, the authors' response does not clarify the point I raised.

Comment 5 (Figure 6G)

The new graph, which is also provided as Figure R3.2 shows that cells expressing RtcB-F wild-type and RtcB-F Y306F behave exactly the same. Nevertheless, the authors state in the text on page 12: "Both WT and Y306F RtcB-Flag rescued XBP1 mRNA splicing in PTP1B^{-/-} MEFs. Importantly, quantification of these results revealed a better rescue of the Y306F RtcB-Flag ...(Fig. 6F, G)." In my view, this contradicts the data.

Comment 6 (Figure 2F)

In my opinion, it is still not clear if RtcB is directly phosphorylated by c-Abl and I agree with reviewer 2 (comment 4) that in vitro kinase assays with purified RtcB variants would need to be done to make the claim that c-Abl phosphorylates RtcB. This is particularly important as the in vitro assay shown in Figure S2D, which the authors use to claim that c-Abl directly phosphorylates RtcB, is inconclusive. The band marked with a black arrowhead in the upper panel of Figure S2D is also present in the first lane, i.e. it could reflect c-Abl autophosphorylation rather than phosphorylation of RtcB by c-Abl.

Minor points

Comment 13 (Figure 7A)

In contrast to what the authors state in the point-by-point response, Figure 7A has not been changed at all.

Referee #1 Review

Report for Author:

When unfolded proteins accumulate in the endoplasmic reticulum (ER stress), cytoprotective mechanism called the ER stress response is activated to cope with it. IRE1 is one of the sensor molecules that sense ER stress. Upon ER stress, IRE1 and RtcB convert XBP1 pre-mRNA into mature mRNA, from which the active transcription factor pXBP1(S) is translated and induces transcription of the ERAD gene, which promotes cell survival. IRE1 cleaves mRNA of various genes by a mechanism called RIDD, resulting in cell death. However, the mechanism controlling the decision to survive or die was unknown.

In this study, the authors found that PTP1B is a target of RIDD and that knocking out the expression of PTP1B reduces the splicing of XBP1 pre-mRNA. Furthermore, they found that RtcB is phosphorylated and dephosphorylated at Y306/316/475 by c-Abl and PTP1B, respectively. The phosphorylated RtcB dissociates from IRE1, resulting in the suppression of XBP1 splicing. Thus, the authors conclude that stress-induced tyrosine phosphorylation of RtcB suppresses its activity, resulting in reduction of XBP1 splicing and finally causing cell death. The reviewers believe that the concept of the manuscript is very interesting and suitable for publication in the journal with revision.

<Critique>

It should be shown that phosphorylation of RtcB at Y306/316/475 is increased in response to ER stress. Furthermore, they should show that PTP1B expression is decreased by ER stress. This is essential for their conclusion. If the phosphorylation of

RtcB is not altered by ER stress, it means that the phosphorylation of RtcB does not regulate the decision of life and death, and the manuscript should be published in another specific journal.

Referee #2 Review

Report for Author:

The purpose of this work is to decipher the mechanisms by which IRE1 and RtcB trigger adaptive or death signals upon ER stress. IRE1 activity balances between life and death decision. IRE1 degrades RNAs through RIDD, which trigger cell death. On the other hand, IRE1 catalyzes XBP1 mRNA splicing together with RtcB. The switch is dependent on the magnitude and length of ER stress. The authors used two independent screens: i. siRNA library against ER genes to identify XBP1s positive and negative regulators upon ER stress; ii. In vitro IRE1 mRNA cleavage assay to identify potential RIDD targets. Intersecting between the siRNA library experiment and the cleavage assay resulted in XBP1s regulators that could be also subjected to RIDD. The results from the two screens identified PTP1B as a tyrosine phosphatase that when dephosphorylates RtcB (Y306 residue) allows the IRE1/RtcB complex to efficiently splice XBP1, thus exerting adaptive UPR. In contrast, using computational modelling as well as biochemical methods, the authors identified c-Abl as the main kinase that phosphorylates three tyrosines in RtcB, which impair the IRE1/RtcB complex and favor RIDD (terminal UPR). The authors concluded that tyrosine phosphorylation dependent regulation of RtcB interaction with IRE1 can confer a mechanism to balance between XBP1 mRNA splicing (adaptive UPR) to RIDD (terminal UPR). Moreover, these results support a dynamic nature of UPRosome that depends of its components can determine the outcome of the UPR.

Overall, this study reveals an interesting and important mechanism by which IRE1/RtcB triggers adaptive versus death signaling upon stress and highly supports the existence of a dynamic multicomplex. A mechanism that was so far ill-defined. Deciphering the mechanism is very important in the understanding of life and death decisions upon stress and can contribute to understand the nature of different UPR related diseases.

The paper is well written. The strength of this paper is its high quality and novel screens that reveal the function of PTP1B in the UPR. In addition, molecular dynamics simulations are well described and contribute to the understanding of the molecular dynamics of RtcB phosphorylation status and its possible mechanism. Other diverse biochemical methods further support the revealed mechanism. However, that are several points that need to be addressed.

Figure 2A and 2D: what is the nature of the band in the control cells after IP? Is it heavy chain? If this is the case, why isn't it equal in all samples including in the Flag-RtcB? Also in both section of this figure there is a problem in loading control which is far from equal. This makes the conclusion about the total phosphorylation state in the cells less convincing.

Figure 2C: The authors showed that RtcB interacts with PTP1B. They nicely took advantage of PTP1B^{+/+} MEFs versus PTP1B^{-/-} MEFs and showed that phosphorylated form of RtcB can be found in IP experiments in PTP1B^{-/-} cells. In the described figure legend, it is written "PTP1B^{+/+} or PTP1B^{-/-} MEFS...treated with 15uM BPv(phen) for two hours". If so, the experiment should be done without BPv(phen) otherwise by using BPv(phen), one should expect to also see the phosphorylated form of RtcB in wt cells as well (as in figure 2A). Please explain why the phosphatase inhibitor was added.

Figure S2C: Endogenous RtcB IP was performed looking at the phosphorylation state of RtcB as well as general phosphorylation state after 15uM BPv(phen) treatment. The authors should include a control (cells without treatment) in the same way as they did for the exogenous RtcB-Flag (S2B).

Figure 2D and Figure S2D: The authors claim that c-Abl is the major tyrosine kinase that phosphorylates RtcB. First by checking different kinase inhibitors only Dasatinib showed reduction in RtcB phosphorylation (Figure 2D). As Dasatinib is not exclusively c-Abl inhibitor, the authors performed an in vitro kinase assay using recombinant proteins and showed that GST-RtcB is phosphorylated by His-C-Abl (figure S2D). This assay is more valuable and specific than use of the different kinase inhibitors and should be moved to figure 2 instead of the Figure2D that can be moved to supplementary figure 2 instead. Following this experiments, the authors performed MS as well as used molecular dynamics simulations and nicely describe three tyrosine residues (Y306, Y316 and Y475), which they claim through all the manuscript that these sites are C-Abl phosphorylation site. Because there are no direct biochemical experiments that showed this, I find this less than conclusive. To show directly that these sites are true c-Abl Phosphorylation sites the authors should repeat the kinase assay with the mutant GST-Y306F, GST-Y316F, GST-Y475F as well as the triple mutant and show a reduction, or no phosphorylation of the mutants. This experiment will strengthen their results.

Figure 4C: "The levels of the phosphorylated RtcB (B) were normalized to RtcB protein levels". Looking at the results of the IP and phosphorylation levels in figure 4B and comparing to the quantification in C, the quantification should be normalized to the amount of the protein that was IPed and not to the input. There are clear differences between the efficiency of the IP in the different mutations which will results in different state of phosphorylation. Normalizing to input will not show these important differences. After re-quantification please change the conclusion to this section in the results ("decrease in tyrosine phosphorylation correlating with the number of Y to F conversion) accordingly.

Figure 4D,E: The authors measured the half-life of the different mutants using CHX assay. Because this is exogenous plasmids that were transfected separately to the cells each mutant should have its own loading control, because those are different transfected cells.

Figure 7B and Figure S8, S9: The authors examined necrosis and apoptotic death decisions upon RtcB Y306 phosphorylation using FACS analysis after Doxorubicin or etoposide treatment and staining the cells with 7AAD and Annexin V. All the

quantification taking into consideration only % cells in Q1 but from the plots there is a clear shift also to the area of Q2. Why didn't the authors also account for Q1+Q2?

Additional minor comments

Details for the in vitro Ire cleavage screen should be provided rather than being cited (indeed even in the cited paper, the description of the screen is not adequate).

Where is the site of Ire1 cleavage of PTP1B?

Referee #3 Review

Report for Author:

The authors investigate how activation of the unfolded protein response (UPR) by ER stress can lead to either stress adaptation or cell death. Specifically, they study which mechanisms control the two alternative activities of the ER stress sensor and UPR signal transducer IRE1, namely (1) cleavage of the XBP1 mRNA to enable ligation of the resulting cleavage fragments by the tRNA ligase RtcB and subsequent production of the active UPR transcription factor XBP1s for cell adaptation, and (2) cleavage of various mRNAs as part of Regulated IRE1-Dependent Decay (RIDD), which may eventually lead to cell death. Starting from the analysis of an siRNA screen for modulators of XBP1 mRNA splicing and an in vitro assay for the identification of RIDD targets, the authors arrive at an attractive model: they propose that the phosphorylation status of RtcB regulates whether IRE1 preferentially cleaves XBP1 mRNA or carries out RIDD. They suggest that RtcB phosphorylation status is determined by the kinase c-Abl and the phosphatase PTP1B and that the PTP1B mRNA is itself a RIDD substrate. Hence, degradation of the PTP1B mRNA through RIDD would inhibit XBP1s production as a result of RtcB hyperphosphorylation. This would tilt the balance away from XBP1 splicing, and thus away from an adaptive UPR, and towards apoptosis as part of a terminal UPR.

This is very interesting work that combines screens, biochemical follow-up work and molecular modeling (the latter of which I'm not qualified to judge). There are small issues throughout (see minor points below). Most importantly, however, the idea that PTP1B-mediated dephosphorylation of residue Y306 in RtcB is important for promoting IRE1-mediated cleavage of XBP1 mRNA in cells needs stronger experimental support to be convincing (see major points 1 to 3, below). This is a key element of the authors' model and needs to be clarified. Furthermore, the claim that c-Abl is the kinase responsible for RtcB phosphorylation in cells appears to be insufficiently supported by evidence presented in this study (see major point 4, below).

If these and the more minor issues can be clarified, this study would certainly be of considerable interest to many in the area of ER stress responses.

Major points:

1. Figure 1E shows that the fold induction of XBP1 splicing upon ER stress is reduced in the absence of PTP1B. The data for WT and PTP1B^{-/-} cells appear to be normalized to the respective untreated samples so that basal XBP1 splicing is set to 1 for both strains. However, Figure S1H indicates that basal XBP1 splicing in PTP1B^{-/-} cells is about 50% higher than in the WT. With this in mind, Figure 1E would indicate that PTP1B knockout causes constitutive XBP1 splicing and that the levels of tunicamycin-induced XBP1 splicing in WT and PTP1B^{-/-} cells are similar. This would be inconsistent with the authors' model, which predicts that PTP1B knockout should, if anything, give reduced basal XBP1 splicing due to RtcB hyperphosphorylation and should reduce XBP1 splicing upon ER stress. To clarify which interpretation is correct, please normalize all of the data in Figure 1E to untreated WT cells.

2. Figures 5A-F are key to the claim that dephosphorylation of Y306 of RtcB promotes XBP1 splicing by IRE1. However, there are several issues with these panels that render this claim unconvincing. First, there are no error bars in Figures 5A and 5D, as noted by the authors in the figure legend. If the panels become too crowded by error bars, it might help to just plot the 8 h time point as a bar graph. Second, the Western blot in 5C, which is used to determine the abundance of the different RtcB variants, appears to contain several saturated bands, both for the anti-FLAG and the anti-actin antibody, making quantification of the blot problematic. Third, Figure 5B indicates that the specific activities of RtcB Y314F and Y475F are the same. However, Figure 5A shows that XBP1 splicing at the relevant 8 h time point is roughly the same for the two mutants and Figure 5C shows that the abundance of RtcB Y475F is much lower. Hence, the specific activity of RtcB Y475F should be considerably higher than that of RtcB Y314F. Fourth, in Figure 5D, all RtcB variants apart from the wt show levels of XBP1 splicing that are below the levels in the parental cells. Thus, the double, triple and H428A mutants are all inactive or even dominant negative. It therefore appears unhelpful to calculate specific activities for them (Figure 5E). In addition, even if the specific activities were meaningful, it would be difficult to understand why the Y306F mutant shows higher activity than the WT (Figure 5A), the Y316F mutant is as active as the WT (Figure 5A) but the Y306F, Y316F double mutant has less activity than the WT (Figure 5E).

3. The authors conclude that RtcB Y306F rescues XBP1 mRNA splicing in PTP1B^{-/-} cells more robustly than RtcB WT (Figures 6F and G). However, the plot on the right of Figure 6G notwithstanding, the data look as if the kinetics of XBP1 splicing in cells expressing RtcB WT and Y306F are the same. To clarify, please plot % XBP1s in PTP1B^{-/-} cells expressing no RtcB-F, RtcB-F wt and RtcB-F Y306F in the same plot (the three red lines in Figure 6G).

4. The claim that c-Abl is the kinase responsible for RtcB phosphorylation appears to rest on the use of the tyrosine kinase inhibitor dasatinib in Figure 2D, E and the fact that c-Abl is able to phosphorylate RtcB in vitro (Figure S2D). However, given that dasatinib is not a specific c-Abl inhibitor, it remains unclear which tyrosine kinase(s) phosphorylate(s) RtcB in cells. This claim should therefore either be backed up by more direct evidence or toned down.

Minor points:

1. Relevant papers to include in the introduction are Li et al., eLife 2021 (PMID 33904404) and Le Thomas et al., biorxiv 2021 (<https://doi.org/10.1101/2021.03.16.435520>).
2. p5, first paragraph: "We identified 23 positive and 32 negative regulators (out of a siRNA library targeting >300 hits) including ...". This should read: "... out of an siRNA library targeting >300 genes."
3. p7, first line: "... showed that RtcB-FLAG exhibited increased tyrosine phosphorylation". The word 'increased' suggests that phosphorylation is higher than under some other condition, but it is not clear what other condition that could be.
4. p7, second line and Figure 2A: In the IP samples of the control (CTL) there are bands at about 55 kDa detected by both the anti-FLAG and the anti-pY antibodies. If these are unspecific bands, please mark them as such.
5. p7, third line and Figure S2C: The IP of endogenous RtcB shows that the major band detected with the anti-RtcB antibody has much lower electrophoretic mobility than the minor band at the expected molecular mass of 55 kDa. This appears to be in contrast to exogenous RtcB-FLAG, although the higher molecular weight range is not shown in Figure 2A. Could the authors please comment?
6. p7, second paragraph and Figure 2C: In the IP samples probed with the anti-pY and the anti-FLAG antibodies, there are bands at 55 kDa in both the control and the RtcB-F lanes. If these are non-specific bands, please mark them as such.
7. p7, second paragraph and Figure 2C: The input samples from PTPB1^{+/+} cells probed with the anti-pY antibody shows a band at 55 kDa that is absent in the samples from PTPB1^{-/-} cells. Is this the result of sequential probing of the membrane with the anti-PTP1B and the anti-pY antibodies? If so, please indicate this, otherwise the anti-pY blot is confusing.
8. p9 and Figure 4B: What is the very prominent band that is precipitated by the anti-FLAG antibody from control cells not expressing RtcB-FLAG, runs at 55 kDa, is recognized by the anti-pY antibody and is much less prominent in cells expressing RtcB-FLAG? Please comment.
9. p10 and Figure 5B: Please indicate what the asterisks above the bar for Y306F and the horizontal line indicate. Which sample was tested against which?
10. p10 and Figure 5E: Same issue - which statistical comparisons do the asterisks represent?
11. p12 and Figure 6E: Has this experiment only been done once? It is an important piece of data and replicates would therefore be needed.
12. p12 and Figure 6F, samples PTPB1^{+/+} cells: the label '24' is underneath the label '0'.
13. p12 and Figure 7A: the y axis plots '% cells necrosis' with values going up to over 1000. Also, the rightmost bracket appears to give a statistical comparison between the sample 'wt siCTL' and 'Y306F siRTCB'. What is the meaning of this comparison?
14. p13, top and p14, bottom: on p13, the authors state that 'apoptosis detection was not obvious [...] thus indicating that the cell death mechanisms triggered in our experiments might not be dependent on the apoptotic cascade'. In the discussion, they then describe the order of events in their proposed model and conclude that 'In parallel c-Abl is recruited to the ER membrane where it phosphorylates RtcB ... enhancing RIDD ..., hence resulting in terminal UPR'. However, in the introduction, the authors clearly equate terminal UPR with apoptosis by saying '... when the stress cannot be resolved, the UPR triggers apoptosis, which is referred to as terminal UPR' (p3, first paragraph). This is confusing.

Authors' Response to Reviewers

Point-by-point response to the reviewers comments

Referee #1:

When unfolded proteins accumulate in the endoplasmic reticulum (ER stress), cytoprotective mechanism called the ER stress response is activated to cope with it. IRE1 is one of the sensor molecules that sense ER stress. Upon ER stress, IRE1 and RtcB convert XBP1 pre-mRNA into mature mRNA, from which the active transcription factor pXBP1(S) is translated and induces transcription of the ERAD gene, which promotes cell survival. IRE1 cleaves mRNA of various genes by a mechanism called RIDD, resulting in cell death. However, the mechanism controlling the decision to survive or die was unknown. In this study, the authors found that PTP1B is a target of RIDD and that knocking out the expression of PTP1B reduces the splicing of XBP1 pre-mRNA. Furthermore, they found that

RtcB is phosphorylated and dephosphorylated at Y306/316/475 by c-Abl and PTP1B, respectively. The phosphorylated RtcB dissociates from IRE1, resulting in the suppression of XBP1 splicing. Thus, the authors conclude that stress-induced tyrosine phosphorylation of RtcB suppresses its activity, resulting in reduction of XBP1 splicing and finally causing cell death. The reviewers believe that the concept of the manuscript is very interesting and suitable for publication in the journal with revision.

We thank the reviewer for this positive evaluation of our work

R1 comment #1 - It should be shown that phosphorylation of RtcB at Y306/316/475 is increased in response to ER stress.

*To address this point we have added new experimental data in **Fig.2F-G** and in **Fig.S6A**, regarding the evaluation of RtcB tyrosine phosphorylation for RtcBWT + TM and under ABL KD (**Fig.2F-G**) and for RtcBY306F/Y316F/Y475F + TM and in ABL KD (**Fig.S6A**). The corresponding text can be found on p7 and 9 of the revised manuscript. From these experiments we conclude that global tyrosine phosphorylation RtcB-WT increased upon TM treatment and decreased in cABL knocked-down cells (**Fig.2F-G**). This supports our finding that regulation by pY of RtcB is an ER stress-related event. Tyrosine phosphorylation of RtcB mutants appeared to be less affected than that of RtcB-WT(**Fig.S6A**). This could be an indication that the phosphorylation events on single tyrosine residues studied herein are interconnected and might regulate each other. Therefore, it might be difficult to expect significant changes on general RtcB tyrosine phosphorylation for mutants carrying a phosphoablating mutation on single tyrosines. In order to solve this, we have launched the development of immunological tools selectively detecting pY of RtcB (ex. Antibodies against P-Y306, P-Y316) as illustrated for P-Y475 in the revised version of the manuscript (the characteristics of these antibodies are presented in **Figure S3** of the revised manuscript - also see the response to reviewer#2). Of note, tyrosine phosphorylation monitored for RtcBY306F upon tunicamycin-induced ER stress exhibited an opposite behaviour compared to that of RtcBWT. The Y475F RtcB mutant appeared to be the least affected one, supporting our observation that Y475 might be the most abundant pY in RtcB upon ER stress.*

R1 comment #2 - Furthermore, they should show that PTP1B expression is decreased by ER stress. This is essential for their conclusion. If the phosphorylation of RtcB is not altered by ER stress, it means that the phosphorylation of RtcB does not regulate the decision of life and death, and the manuscript should be published in another specific journal.

*To address these points we have added new experimental data in **Fig. S1J** and **Fig. S7E**, respectively. As observed for mRNA levels, PTP1B protein expression levels decreased upon Tunicamycin-induced ER stress (**Fig.S1J**). Moreover, mRNA levels of the RIDD targets PER1 and SCARA3 (used already in Fig.5G) showed a significant decrease under conditions of Tunicamycin-induced ER stress (**Fig.S7E**). The corresponding text can be found on p6 and 10 of the revised manuscript.*

Collectively, our data show that RtcB tyrosine phosphorylation is regulated upon ER stress, and that the expression of the tyrosine phosphatase involved in such

regulation is altered under the same conditions, thus pointing towards an integrated regulation of RtcB tyrosine phosphorylation.

Referee #2:

The purpose of this work is to decipher the mechanisms by which IRE1 and RtcB trigger adaptive or death signals upon ER stress. IRE1 activity balances between life and death decision. IRE1 degrades RNAs through RIDD, which trigger cell death. On the other hand, IRE1 catalyzes XBP1 mRNA splicing together with RtcB. The switch is dependent on the magnitude and length of ER stress. The authors used two independent screens: i. siRNA library against ER genes to identify XBP1s positive and negative regulators upon ER stress; ii. In vitro IRE1 mRNA cleavage assay to identify potential RIDD targets. Intersecting between the siRNA library experiment and the cleavage assay resulted in XBP1s regulators that could be also subjected to RIDD. The results from the two screens identified PTP1B as a tyrosine phosphatase that when dephosphorylates RtcB (Y306 residue) allows the IRE1/RtcB complex to efficiently splice XBP1, thus exerting adaptive UPR. In contrast, using computational modelling as well as biochemical methods, the authors identified c-Abl as the main kinase that phosphorylates three tyrosines in RtcB, which impair the IRE1/RtcB complex and favor RIDD (terminal UPR). The authors concluded that tyrosine phosphorylation dependent regulation of RtcB interaction with IRE1 can confer a mechanism to balance between XBP1 mRNA splicing (adaptive UPR) to RIDD (terminal UPR). Moreover, these results support a dynamic nature of UPRosome that depends of its components can determine the outcome of the UPR. Overall, this study reveals an interesting and important mechanism by which IRE1/RtcB triggers adaptive versus death signaling upon stress and highly supports the existence of a dynamic multicomplex. A mechanism that was so far ill-defined. Deciphering the mechanism is very important in the understanding of life and death decisions upon stress and can contribute to understand the nature of different UPR related diseases. The paper is well written. The strength of this paper is its high quality and novel screens that reveal the function of PTP1B in the UPR. In addition, molecular dynamics simulations are well described and contribute to the understanding of the molecular dynamics of RtcB phosphorylation status and its possible mechanism. Other diverse biochemical methods further support the revealed mechanism. However, that are several points that need to be addressed.

We thank the reviewer for this positive and constructive evaluation of our work.

Referee #2 – comment 1: Figure 2A and 2D: what is the nature of the band in the control cells after IP? Is it heavy chain? If this is the case, why isn't it equal in all samples including in the Flag-RtcB? Also in both section of this figure there is a problem in loading control which is far from equal. This makes the conclusion about the total phosphorylation state in the cells less convincing.

We extensively revised Fig. 2 to address this reviewer's first comment. As such we relied on the fact that both in phosphoproteomics databases and in our own proteomics experiments, the most represented phosphotyrosine-containing peptide was the peptide containing Y475. In an attempt to use this phosphopeptide as a proxy of RtcB tyrosine phosphorylation, we developed an immunological tool for the selective detection of pY of

RtcB at Y475. The description of this tool (anti pY475) is provided in the revised Fig. S3 and the corresponding text can be found on p7 in the Results section and on p18 in the Methods section of the revised manuscript. In Fig. S3, a schematic representation of antibody design, pY475 levels of RtcB WT and Y475F in HEK293T transiently transfected and HeLa stably transfected. In HeLa cells, Flag-RtcB was immunoprecipitated and the immune complexes immunoblotted with anti- pY475 antibodies. As such we used this tool to replace Fig. 2D with a new one. Indeed, the initial experiments carried out with a general anti-pY antibody were repeated by using the pY475 antibody (controls are provided in revised Fig. S3B-C). The results confirmed what was initially suggested with a decreased phosphorylation of RtcB Y475 upon treatment with Dasatinib, an inhibitor of c-ABL and SRC.

Referee #2 – comment 2: Figure 2C: The authors showed that RtcB interacts with PTP1B. They nicely took advantage of PTP1B^{+/+} MEFs versus PTP1B^{-/-} MEFs and showed that phosphorylated form of RtcB can be found in IP experiments in PTP1B^{-/-} cells. In the described figure legend, it is written "PTP1B^{+/+} or PTP1B^{-/-} MEFS...treated with 15uM BPv(phen) for two hours". If so, the experiment should be done without BPv(phen) otherwise by using BPv(phen), one should expect to also see the phosphorylated form of RtcB in wt cells as well (as in figure 2A). Please explain why the phosphatase inhibitor was added.

This is true, the experiments were performed only in the presence of bpV(phen), indeed the addition of the phosphatase inhibitor allowed for robust detection (consistent) of endogenous RtcB tyrosine phosphorylation. PTPase inhibition was necessary in the case of endogenous RtcB probably because the protein is not abundantly expressed (in contrast to Flag-RtcB) and also probably because RtcB tyrosine phosphorylation might be relatively transient. We are conscious that bpV(phen) treatment might induce experimental biases and as such we have added a statement in the results section on p6 of the revised manuscript. Regarding both PTP1B^{+/+} and PTP1B^{-/-} cells we observed a slight band in PTP1B^{+/+} cells already which could be indicative that in MEF cells RtcB tyrosine phosphorylation might not be that strong compared to the one measured in HEK or HeLa cells. What is important, however, is that the absence of PTP1B (PTP1B^{-/-} cells) alone greatly increased Tyr phosphorylation level of RtcB, pointing to its main role in the de-phosphorylation of RtcB compared to other candidate Tyr phosphatases.

Referee #2 – comment 3: Figure S2C: Endogenous RtcB IP was performed looking at the phosphorylation state of RtcB as well as general phosphorylation state after 15uM BPv(phen) treatment. The authors should include a control (cells without treatment) in the same way as they did for the exogenous RtcB-Flag (S2B).

The reason why the experiment in Fig. S2B was performed was to test whether inhibiting protein tyrosine phosphatases (thereby increasing tyrosine phosphorylation) would help on the detection of phosphorylated exogenous RtcB which represents the aim of part of the experiments included in the manuscript. Unfortunately, the blot shows that it is not possible to detect FLAG-RtcB from total lysates but clearly shows how the treatment increases the total tyrosine phosphorylation. Based on this observation we included the treatment with BPv(phen) in all our experiments. In order to detect pFLAG-RtcB, we performed a FLAG immunoprecipitation to magnify the detection of phosphotyrosine-RtcB which is otherwise undetectable. This is in agreement with the response to the previous comment as follows: "This is true, the experiments were performed only in the presence of bpV(phen), indeed the

addition of the phosphatase inhibitor allowed for robust detection (consistent) of endogenous RtcB tyrosine phosphorylation. PTPase inhibition was necessary in the case of endogenous RtcB probably because the protein is not abundantly expressed and also probably because RtcB tyrosine phosphorylation might be relatively transient.”

Referee #2 – comment 4: Figure 2D and Figure S2D: The authors claim that c-Abl is the major tyrosine kinase that phosphorylates RtcB. First by checking different kinase inhibitors only Dasatinib showed reduction in RtcB phosphorylation (Figure 2D). As Dasatinib is not exclusively c-Abl inhibitor, the authors performed an in vitro kinase assay using recombinant proteins and showed that GST-RtcB is phosphorylated by His-C-Abl (figure S2D). This assay is more valuable and specific than use of the different kinase inhibitors and should be moved to figure 2 instead of the Figure2D that can be moved to supplementary figure 2 instead. Following this experiments, the authors performed MS as well as used molecular dynamics simulations and nicely describe three tyrosine residues (Y306, Y316 and Y475), which they claim through all the manuscript that these sites are C-Abl phosphorylation site. Because there are no direct biochemical experiments that showed this, I find this less than conclusive. To show directly that these sites are true c-Abl Phosphorylation sites the authors should repeat the kinase assay with the mutant GST-Y306F, GST-Y316F, GST-Y475F as well as the triple mutant and show a reduction, or no phosphorylation of the mutants. This experiment will strengthen their results.

*In order to solve this issue we opted to immunoprecipitate RtcB wt and mutants and monitor the immunoprecipitated proteins tyrosine phosphorylation in c-ABL-depleted cells (using a siRNA-based knocked down approach). We did not select a kinase assay that would have required to purify RtcB mutants (in addition to create other biases regarding the stoichiometry between kinase and substrate). The results obtained in these experiments were in line with what was suggested by the results shown in **Fig. 2D** and which showed decreased tyrosine phosphorylation of RtcB upon Dasatinib treatment. In the new experiments performed in c-ABL-depleted cells, RtcB tyrosine phosphorylation was attenuated. These results are presented in **Fig. 2F-G** in the revised manuscript and the corresponding text can be found on p7. Moreover, these results showed that the Y306F mutant exhibited a behaviour similar to that of RtcBWT. Interestingly the Y316F RtcB mutant showed an increased tyrosine phosphorylation which was the opposite of what was observed for RtcB-WT and the Y306F mutant. This might be indicative of the implication of other kinases phosphorylating these residues outside ABL (e.g. recent data in the laboratory indicate that SRC could phosphorylate RtcB on Y5). Furthermore, the Y475F RtcB mutant appeared to be the least affected in those experiments. This result supported our initial finding that the Y475 might be the best c-ABL substrate (as reflected by its abundance) in RtcB upon ER stress (**Fig. S6A**). Collectively our results confirm that RtcB contains tyrosine residues substrates of c-ABL kinase activity. Moreover, they may indicate as well that the sequence of tyrosine phosphorylation on RtcB might impact on the relative phosphorylation of these different tyrosines. A sentence reporting this has been added in the revised version of the manuscript on p9.*

Referee #2 – comment 5: Figure 4C: "The levels of the phosphorylated RtcB (B) were normalized to RtcB protein levels". Looking at the results of the IP and phosphorylation

levels in figure 4B and comparing to the quantification in C, the quantification should be normalized to the amount of the protein that was IPed and not to the input. There are clear differences between the efficiency of the IP in the different mutations which will result in different state of phosphorylation. Normalizing to input will not show these important differences. After re-quantification please change the conclusion to this section in the results ("decrease in tyrosine phosphorylation correlating with the number of Y to F conversion) accordingly.

The experiment was modified as requested and the quantification performed for Fig. 4B. The conclusion drawn from the results was modified in the revised version of the manuscript and can be seen on p9.

Referee #2 – comment 6: Figure 4D,E: The authors measured the half-life of the different mutants using CHX assay. Because this is exogenous plasmids that were transfected separately to the cells each mutant should have its own loading control, because those are different transfected cells.

As pointed out by this reviewer, all the quantifications were performed with the experimental condition loading control (in this case VCP which is a known long-lived protein). As seen on

the left, a representative western blot of RtcB (and mutants) expression upon cycloheximide treatment (with variable half-lives) and the corresponding expression of VCP. This representative blot could possibly be shown as supplemental material (Fig.S7A).

Figure R2.1 – HeLa cells expressing or not various Flag tagged versions of RtcB (WT, single mutants, double or triple mutants) were treated for 0, 1, 2, 4, 8h with 10µg/ml cycloheximide. Cell lysates were harvested and the expression of Flag-RtcB, endogenous RtcB or VCP evaluated using western blot with the corresponding antibodies.

Referee #2 – comment 7: Figure 7B and Figure S8, S9: The authors examined necrosis and apoptotic death decisions upon RtcB Y306 phosphorylation using FACS analysis after Doxorubicin or etoposide treatment and staining the cells with 7AAD and Annexin V. All the quantification taking into consideration only % cells in Q1 but from the plots there is a clear shift also to the area of Q2. Why didn't the authors also account for Q1+Q2?

We agree with this reviewer's remark. Indeed the Q1 fraction is indicative of necrotic cells whereas Q2 and Q4 relate to early and late apoptotic cells, respectively. We focused on the Q1 fraction as it was the one that was affected the most reproducibly by the treatments to which the cells were exposed to and by the phosphorylation of RtcB. The Q2 fraction, even though showing a shift on the presented plots, did not reach statistical significance. This indicated that favoring XBP1 mRNA splicing under DNA damage stress conditions is not protective but rather precipitates the cells into a death path that might not require apoptotic-dependent mechanisms. This interpretation of the data has now been added in the revised version of the manuscript on p 13.

Additional minor comments

Details for the in vitro Ire cleavage screen should be provided rather than being cited (indeed even in the cited paper, the description of the screen is not adequate).

The detailed protocol for the in vitro cleavage screen can be found in Lhomond et al. 2018 as indicated in the manuscript. However, for the sake of precision, we have extended the details of protocol in Materials and methods on p17 of the revised manuscript.

Where is the site of Ire1 cleavage of PTP1B?

We have not identified the IRE1-mediated cleavage site on PTP1B mRNA yet. Based on our own experience (unpublished results) as well as on the recent preprint by Le Thomas and colleagues (doi: <https://doi.org/10.1101/2021.03.16.435520>), we searched for potential cleavage site on PTP1B mRNA based on consensus sequences and then evaluated whether these sites localized in P-loop structures. This analysis unveiled 7 potential sites of which 4 were canonical (CUGCAG) and 3 were non-canonical. Structural analysis of the regions containing these sites so far revealed that one of the non canonical site was comprised within a P-loop only. As such additional experimental work is currently ongoing to characterize IRE1-mediated cleavage of PTP1B mRNA.

Referee #3:

The authors investigate how activation of the unfolded protein response (UPR) by ER stress can lead to either stress adaptation or cell death. Specifically, they study which mechanisms control the two alternative activities of the ER stress sensor and UPR signal transducer IRE1, namely (1) cleavage of the XBP1 mRNA to enable ligation of the resulting cleavage fragments by the tRNA ligase RtcB and subsequent production of the active UPR transcription factor XBP1s for cell adaptation, and (2) cleavage of various mRNAs as part of Regulated IRE1-Dependent Decay (RIDD), which may eventually lead to cell death. Starting from the analysis of an siRNA screen for modulators of XBP1 mRNA splicing and an in vitro assay for the identification of RIDD targets, the authors arrive at an attractive model: they propose that the phosphorylation status of RtcB regulates whether IRE1 preferentially cleaves XBP1 mRNA or carries out RIDD. They suggest that RtcB phosphorylation status is determined by the kinase c-Abl and the phosphatase PTP1B and that the PTP1B mRNA is itself a RIDD substrate. Hence, degradation of the PTP1B mRNA through RIDD would inhibit XBP1s production as a result of RtcB hyperphosphorylation. This would tilt the balance away from XBP1 splicing, and thus away from an adaptive UPR, and towards apoptosis as part of a terminal UPR. This is very interesting work that combines screens, biochemical follow-up work and molecular modeling (the latter of which I'm not qualified to judge). There are small issues throughout (see minor points below). Most importantly, however, the idea that PTP1B-mediated dephosphorylation of residue Y306 in RtcB is important for promoting IRE1-mediated cleavage of XBP1 mRNA in cells needs stronger experimental support to be convincing (see major points 1 to 3, below). This is a key element of the authors' model and needs to be clarified. Furthermore, the claim that c-Abl is the kinase responsible for RtcB phosphorylation in cells appears to be insufficiently supported by evidence presented in this study (see major point 4, below).

If these and the more minor issues can be clarified, this study would certainly be of considerable interest to many in the area of ER stress responses.

As for reviewers 1 and 2, we thank reviewer 3 for this positive and constructive evaluation of our work, we hope that our responses and changes made to the revised manuscript will be satisfactory.

Major points:

Referee #3 – comment 1. Figure 1E shows that the fold induction of XBP1 splicing upon ER stress is reduced in the absence of PTP1B. The data for WT and PTP1B^{-/-} cells appear to be normalized to the respective untreated samples so that basal XBP1 splicing is set to 1 for both strains. However, Figure S1H indicates that basal XBP1 splicing in PTP1B^{-/-} cells is about 50% higher than in the WT. With this in mind, Figure 1E would indicate that PTP1B knockout causes constitutive XBP1 splicing and that the levels of tunicamycin-induced XBP1 splicing in WT and PTP1B^{-/-} cells are similar. This would be inconsistent with the authors' model, which predicts that PTP1B knockout should, if anything, give reduced basal XBP1 splicing due to RtcB hyperphosphorylation and should reduce XBP1 splicing upon ER stress. To clarify which interpretation is correct, please normalize all of the data in Figure 1E to untreated **WT** cells.

After normalizing all of the data in Fig. 1E to untreated PTP1B^{+/+} cells, we indeed observe greater XBP1 mRNA splicing levels in PTP1B^{-/-} cells under basal conditions which is in agreement with Fig. S1H. However, this is reversed upon treatment with tunicamycin for 3 and 6 hours, meaning that PTP1B^{-/-} cells have now lower XBP1 splicing levels coming in agreement with what was reported originally with Fig. 1E.

Referee #3 – comment 2. Figures 5A-F are key to the claim that dephosphorylation of Y306 of RtcB promotes XBP1 splicing by IRE1. However, there are several issues with these panels that render this claim unconvincing. First, there are no error bars in Figures 5A and 5D, as noted by the authors in the figure legend. If the panels become too crowded by error bars, it might help to just plot the 8 h time point as a bar graph.

The panels were indeed becoming too crowded when adding the error bars, but to address this reviewer's comment we have inserted graphs containing error bars on Fig. 5 A and D. We have plotted the 8h time point as a bar graph which is presented below (Figure R3.1).

Figure R3.1 – XBP1 mRNA splicing following 8h Tun treatment in cells expressing RtcB mutants or WT

The increased levels of XBP1 splicing of the mutant Y306F compared to the WT RtcB are still visible and they come in agreement with Fig. 5B where a normalization to the corresponding Flag-RtcB protein levels was used.

Referee #3 – comment 3. Second, the Western blot in 5C, which is used to determine the abundance of the different RtcB variants, appears to contain several **saturated** bands, both for the anti-FLAG and the anti-actin antibody, making quantification of the blot problematic.

The blot was replaced by a less saturated image.

Referee #3 – comment 3. Third, Figure 5B indicates that the specific activities of RtcB Y314F and Y475F are the same. However, Figure 5A shows that XBP1 splicing at the relevant 8 h time point is roughly the same for the two mutants and Figure 5C shows that the abundance of RtcB Y475F is much lower. Hence, the specific activity of RtcB Y475F should be considerably higher than that of RtcB Y314F.

Quantification was carried out on 3 independent experiments, and a representative image of the blots is shown in the revised Fig. 5C.

Referee #3 – comment 4. Fourth, in Figure 5D, all RtcB variants apart from the wt show levels of XBP1 splicing that are below the levels in the parental cells. Thus, the double, triple and H428A mutants are all inactive or even dominant negative. It therefore appears unhelpful to calculate specific activities for them (Figure 5E). In addition, even if the specific activities were meaningful, it would be difficult to understand why the Y306F mutant shows higher activity than the WT (Figure 5A), the Y316F mutant is as active as the WT (Figure 5A) but the Y306F, Y316F double mutant has less activity than the WT (Figure 5E).

This suggest a pY-based regulation of RtcB functions that operates not only through single tyrosine phosphorylation but also trough a combination of phosphorylated tyrosine residues. This also highlights the complexity of those event and might suggest a hierarchy in the events in order to reach the desired biological outcome.

Referee #3 – comment 5. The authors conclude that RtcB Y306F rescues XBP1 mRNA splicing in PTP1B^{-/-} cells more robustly than RtcB WT (Figures 6F and G). However, the plot on the right of Figure 6G notwithstanding, the data look as if the kinetics of XBP1 splicing in cells expressing RtcB WT and Y306F are the same. To clarify, please plot % XBP1s in PTPB1^{-/-} cells expressing no RtcB-F, RtcB-F wt and RtcB-F Y306F in the same plot (the three red lines in Figure 6G).

To address this reviewer's point, we aggregated the plots to provide a single graph comprising CTL, Flag-RTCB-WT and Flag-RtcB-Y306F (see below). This graph is shown in the revised Fig. 6G.

Figure R3.2 – revised figure 6G providing on a single graph the comparison of XBP1 mRNA splicing in CTL cells and cells expressing either Flag-RtcB-WT or Flag-RtcB-Y306F.

Referee #3 – comment 6. The claim that c-Abl is the kinase responsible for RtcB phosphorylation appears to rest on the use of the tyrosine kinase inhibitor dasatinib in Figure

2D, E and the fact that c-Abl is able to phosphorylate RtcB in vitro (Figure S2D). However, given that dasatinib is not a specific c-Abl inhibitor, it remains unclear which tyrosine kinase(s) phosphorylate(s) RtcB in cells. This claim should therefore either be backed up by more direct evidence or toned down.

*This reviewer is right, the point was addressed as well in response to **reviewer 2** as follows: "In order to solve this issue we opted to immunoprecipitate RtcB wt and mutants and monitor the immunoprecipitated proteins tyrosine phosphorylation in c-Abl-depleted cells (using a siRNA-based knocked down approach). We did not select a kinase assay that would have required to purify RtcB mutants (in addition to create other biases regarding the stoichiometry between kinase and substrate). The results obtained in these experiments were in line with what was suggested by the results shown in **Fig. 2D** and which showed decreased tyrosine phosphorylation of RtcB upon Dasatinib treatment. In the new experiments performed in c-Abl-depleted cells, RtcB tyrosine phosphorylation was attenuated. These results are presented in **Fig. 2F-G** in the revised manuscript and the corresponding text can be found on p7. Moreover, these results showed that the Y306F mutant exhibited a behaviour similar to that of RtcBWT. Interestingly the Y316F RtcB mutant showed an increased tyrosine phosphorylation which was the opposite of what was observed for RtcBWT and the Y306F mutant. This might be indicative of the implication of other kinases phosphorylating these residues outside Abl (e.g. recent data in the laboratory indicate that SRC could phosphorylate RtcB on Y5). Furthermore, the Y475F RtcB mutant appeared to be the least affected in those experiments. This result supported our initial finding that the Y475 might be the best c-Abl substrate (as reflected by its abundance) in RtcB upon ER stress (**Fig. S6A**). Collectively our results confirm that RtcB contains tyrosine residues substrates of c-Abl kinase activity. Moreover, they may indicate as well that the sequence of tyrosine phosphorylation on RtcB might impact on the relative phosphorylation of these different tyrosines. A sentence reporting this has been added in the revised version of the manuscript on p9."*

Minor points:

1. Relevant papers to include in the introduction are Li et al., eLife 2021 (PMID 33904404) and Le Thomas et al., biorxiv 2021 (<https://doi.org/10.1101/2021.03.16.435520>).

*These articles/preprints were added in the introduction section. Moreover, the discussion section, we commented on the recent preprint by Belyy et al. (doi: <https://doi.org/10.1101/2021.09.29.462487>) and discuss how the model proposed in this manuscript could be in accordance with the model presented in **Fig. 7** of the revised manuscript.*

2. p5, first paragraph: "We identified 23 positive and 32 negative regulators (out of a siRNA library targeting >300 hits) including ...". This should read: ... out of an siRNA library targeting >300 genes.

This was modified as suggested by this reviewer and can be seen on p4 of the revised manuscript.

3. p7, first line: "... showed that RtcB-FLAG exhibited increased tyrosine phosphorylation". The word 'increased' suggests that phosphorylation is higher than under some other condition, but it is not clear what other condition that could be.

The text was modified in order to make this statement more accurate and can be seen on p6 of the revised manuscript.

4. p7, second line and Figure 2A: In the IP samples of the control (CTL) there are bands at about 55 kDa detected by both the anti-FLAG and the anti-pY antibodies. If these are unspecific bands, please mark them as such.

Figure 2A was removed as part of a major restructuring of Figure 2 as indicated in the response to the comments raised by reviewer 1: "We extensively revised Fig. 2 to address this reviewer's first comment. As such we relied on the fact that both in phosphoproteomics databases and in our own proteomics experiments, the most represented phosphotyrosine-containing peptide was the peptide containing Y475. In an attempt to use this phosphopeptide as a proxy of RtcB tyrosine phosphorylation, we developed an immunological tool for the selective detection of pY of RtcB at Y475. The description of this tool (anti pY475) is provided in the revised Fig. S3 and the corresponding text can be found on p7 in the Results section and on p18 in the Methods section of the revised manuscript. In Fig. S3, a schematic representation of antibody design, pY475 levels of RtcB WT and Y475F in HEK293T transiently transfected and HeLa stably transfected. In HeLa cells, Flag-RtcB was immunoprecipitated and the immune complexes immunoblotted with anti- pY475 antibodies. As such we used this tool to replace Fig. 2D with a new one. Indeed, the initial experiments carried out with a general anti-pY antibody were repeated by using the pY475 antibody (controls are provided in revised Fig. S3B-C). The results confirmed what was initially suggested with a decreased phosphorylation of RtcB Y475 upon treatment with Dasatinib, an inhibitor of c-ABL and SRC."

5. p7, third line and Figure S2C: The IP of endogenous RtcB shows that the major band detected with the anti-RtcB antibody has much lower electrophoretic mobility than the minor band at the expected molecular mass of 55 kDa. This appears to be in contrast to exogenous RtcB-FLAG, although the higher molecular weight range is not shown in Figure 2A. Could the authors please comment?

We did not observe such pattern when the experiments were carried out using anti-Flag antibodies to immunoprecipitate Flag-RtcB and use the Flag antibodies for western blot. Based on this observation we could speculate either that the RtcB antibodies might yield non RtcB-specific signals (both in immunoprecipitation and western blotting experiments) or that they pick an alternative variant of RtcB that cannot be detected using recombinant Flag-RtcB. This needs to be further clarified.

6. p7, second paragraph and Figure 2C: In the IP samples probed with the anti-pY and the anti-FLAG antibodies, there are bands at 55 kDa in both the control and the RtcB-F lanes. If these are non-specific bands, please mark them as such.

*This was performed as requested and can be seen on the revised **Fig. 2B** and in the corresponding figure legends.*

7. p7, second paragraph and Figure 2C: The input samples from PTPB1^{+/+} cells probed with the anti-pY antibody shows a band at 55 kDa that is absent in the samples from PTPB1^{-/-} cells. Is this the result of sequential probing of the membrane with the anti-PTP1B and the anti-pY antibodies? If so, please indicate this, otherwise the anti-pY blot is confusing.

The input samples were separately probed with the corresponding antibodies so there was no stripping involved. PTP1B itself can be tyrosine phosphorylated by tyrosine kinases (e.g. IR), so the band seen only in the PTP1B^{+/+} cells just below 55kDa could actually correspond to the PTP1B protein itself (its absence in the PTP1B^{-/-} cells could explain the absence of this Tyr phosphorylated band there).

8. p9 and Figure 4B: What is the very prominent band that is precipitated by the anti-FLAG antibody from control cells not expressing RtcB-FLAG, runs at 55 kDa, is recognized by the anti-pY antibody and is much less prominent in cells expressing RtcB-FLAG? Please comment.

We agree that this figure was misleading and added more repeats to this experiment that led us to replace the actual panel. We believe that the initially observed band might be unspecific as it did not appear in most of the repeats. Moreover, since we use anti-pY antibodies directly conjugated to HRP, we can rule out the presence of Ig heavy chains.

9. p10 and Figure 5B: Please indicate what the asterisks above the bar for Y306F and the horizontal line indicate. Which sample was tested against which?

Each mutant was compared to the wt using ordinary one-way Anova. The horizontal line with the asterisk indicates that there is a significant difference among the groups. This information has been added in the corresponding figure legend in the revised manuscript.

10. p10 and Figure 5E: Same issue - which statistical comparisons do the asterisks represent?

*This was fixed in the revised version of the manuscript (p10 and **Fig. 5E**).*

11. p12 and Figure 6E: Has this experiment only been done once? It is an important piece of data and replicates would therefore be needed.

*Replicates were performed and raw data are shown in the "appended original data file". A statistical analysis was also carried out and is provided in the revised **Fig. 6E**.*

12. p12 and Figure 6F, samples PTPB1^{+/+} cells: the label '24' is underneath the label '0'.

This was fixed in the revised version of the manuscript.

13. p12 and Figure 7A: the y axis plots '% cells necrosis' with values going up to over 1000. Also, the rightmost bracket appears to give a statistical comparison between the sample 'wt siCTL' and 'Y306F siRTCB'. What is the meaning of this comparison?

This were labelling mistakes, we thank the reviewer for picking them up. The graph is now properly annotated in the revised Fig. 7A.

14. p13, top and p14, bottom: on p13, the authors state that 'apoptosis detection was not obvious [...] thus indicating that the cell death mechanisms triggered in our experiments might not be dependent on the apoptotic cascade'. In the discussion, they then describe the order of events in their proposed model and conclude that 'In parallel c-Abl is recruited to the ER membrane where it phosphorylates RtcB ... enhancing RIDD ..., hence resulting in terminal UPR'. However, in the introduction, the authors clearly equate terminal UPR with apoptosis by saying '... when the stress cannot be resolved, the UPR triggers apoptosis, which is referred to as terminal UPR' (p3, first paragraph). This is confusing.

We agree with this reviewer's comment. In order to minimize confusion we modified the introduction section in p3 of the revised manuscript, to change "apoptosis" into "cell death" in light of our recent review on the relationship between terminal ER stress and cell death (McGrath et al. Biochim Biophys Acta Mol Cell Res. 2021 May;1868(6):119001—this reference was added to the revised manuscript). Moreover, to further document our choices of presenting cell death results rather than apoptosis, we responded to a comment from reviewer #2 as follows: "Indeed the Q1 fraction is indicative of necrotic cells whereas Q2 and Q4 relate to early and late apoptotic cells, respectively. We focused on the Q1 fraction as it was the one that was affected the most reproducibly by the treatments to which the cells were exposed to and by the phosphorylation of RtcB. The Q2 fraction, even though showing a shift on the presented plots, did not reach statistical significance. This indicated that favoring XBP1 mRNA splicing under DNA damage stress conditions is not protective but rather precipitates the cells into a death path that might not require apoptotic-dependent mechanisms. This interpretation of the data has now been added in the revised version of the manuscript on p 12".

January 21, 2022

Re: Life Science Alliance manuscript #LSA-2022-01379-T

Dr. Eric Chevet
INSERM U1242
Rennes 35000
FRANCE

Dear Dr. Chevet,

Thank you for submitting your manuscript entitled "Stress-induced tyrosine phosphorylation of RtcB modulates IRE1 activity and signaling outputs" to Life Science Alliance. We invite you to re-submit the manuscript, revised to address the following:

- Address Reviewer 3's concerns, and provide all source data used for quantification.

Thank you for this interesting contribution to Life Science Alliance. We are looking forward to receiving your revised manuscript.

Sincerely,

B. MANUSCRIPT ORGANIZATION AND FORMATTING:

Referee #3

Papaioannou and colleagues have responded to my comments but, unfortunately, have not satisfactorily resolved the issues I viewed as major (see detailed explanation below). While the proposed model remains intriguing, it is not convincingly supported by the evidence presented and I therefore cannot recommend publication in this journal.

Major points

Comment 1 (Figure 1E) - In the re-plotted graph, the difference in Xbp1 splicing between tunicamycin-treated control and PTPB1 knockout cells is very minor (seven-fold versus six-fold above untreated control cells). It is not clear if this effect is meaningful or even statistically significant. The figure legend states that "Data values are the mean {plus minus} SEM of 4 independent experiments. Unpaired t test was applied for the statistical analyses." However, it is unclear what the asterisks shown in Figure 1 E refer to, i.e. it is not indicated which sample was tested against which by a t-test. Also, it is not stated if the t-test used assumed equal or unequal variance, which is important because the data are normalized. Finally, text states that "... this was reversed upon treatment with TM for 3 and 6 hours" but there are no data for the 3 h time point in Figure 1E. Because of these problems, it remains unclear if stress-induced Xbp1 signaling is impaired in PTPB1 knockout cells. Figure S1E-G does not help to clarify the issue because the experiment shown there was done only once.

Response: We are sorry that the statistical comparisons were not clear. As requested, in this revised version we have clarified the meaning of the asterisks in the figure legend and addressed the t-test question (in the figure legends - highlighted in blue). We also modified the panel E in figure 1 in which data are presented as fold-changes in XBP1 mRNA (spliced and total) to emphasize the fact that PTP1B KO cells are less prone to splice than their WT counterparts and included a sentence in a revised version of the manuscript that describes this on p5 of the manuscript (highlighted in blue). This will certainly increase the strength of the observation obtained in PTP1B^{+/+} and PTP1B^{-/-} cells. To further document the issue raised by reviewer 3, we provide the raw data of the triplicates with the statistics that were done and explain how the statistical tests were carried out (**Figure R1**, PRISM-Graphpad screenshot).

Figure R1: PRISM – Graphpad statistical file for the 2 PCR amplicons

Based on this reviewer's comment about the small effects observed, one could hypothesize that the concentration of tunicamycin used in our study was high (10 ug/ml) thereby triggering both XBP1 mRNA splicing and RIDD. Since PTP1B is also a RIDD target, it might be that the actual levels of PTP1B were decreased in PTP1B+/- cells, thus attenuating the effects on XBP1 mRNA splicing (see our model on **Figure 7**). Indeed, when the same experiment was performed with lower concentration of the stressor (50 ng/ml) (see **Figure 6G**, even though the methods used are different: qPCR vs PCR) the effect on XBP1 mRNA splicing when PTP1B was depleted is much stronger. Moreover, we propose that PTP1B regulates XBP1 mRNA splicing through RtcB dephosphorylation but we never claim that this is the only mechanism for RtcB to regulate IRE1. We have clarified this issue in the revised version. For instance, the RtcB interactome (in basal conditions and upon ER stress) has not yet been explored but protein-protein interaction databases (ex. Biogrid) suggest that RtcB may interact with i) structural proteins as Vimentin, Kinesins, etc.. suggesting a potential involvement of RtcB as part of a scaffolding complex able to integrate mechanical stress from outside the cell (or in the cytosol), ii) components of the ufmylation system, iii) components of the translation machinery and NUPR1 which all could be involved in the regulation of IRE1 signaling outputs. The regulation of RtcB interactome by tyrosine phosphorylation is currently under investigation in the laboratory.

Comment 2 (Figure 5A) - The authors re-plotted the data as suggested and show them in the point-by-point response as Figure R3.1. This graph clearly shows that even at the most favorable time point (8 h of tunicamycin treatment) there is no statistically significant difference between cells expressing RtcB WT and RtcB Y306F (second and third bar). The authors do not provide any statistics but judging from the individual data points, the values for cells expressing RtcB WT are approximately 2, 3.8 and 5. The values for the cells expressing RtcB Y306F are approximately 3, 5.7 and 6.5. Testing these data sets against each other with a two-tailed t-test assuming equal variance yields $p > 0.3$. The data therefore do not allow the conclusion that cells expressing RtcB WT and RtcB Y306F are different.

Response: As requested, in the revised version we provide statistical analyses to reinforce our main conclusions. It is very surprising that data presented, for instance in Figure 5A, B, and commented in text and first rebuttal, are not taken into consideration by the reviewer as they may be in contradiction with his/her views. Our conclusions are clearly supported by the results presented in Figure 5B, which forms the basis for the claims made in our manuscript. We apologize that this new data was not explicit for the reviewer and have revised the text accordingly to improve clarity. We provide below the statistical analysis carried out on the data to which the reviewer is referring to (**Figure R2**).

ANOVA results		
1	Table Analyzed	Specific activity @8h when protein level normalized_n=3
2	Data sets analyzed	A-D
3		
4	ANOVA summary	
5	F	10.22
6	P value	0.0041
7	P value summary	**
8	Significant diff. among means (P < 0.05)?	Yes
9	R square	0.7931
10		
11	Brown-Forsythe test	
12	F (DFn, DFd)	1.958 (3, 8)
13	P value	0.1990
14	P value summary	ns
15	Are SDs significantly different (P < 0.05)?	No
16		
17	Bartlett's test	
18	Bartlett's statistic (corrected)	
19	P value	
20	P value summary	
21	Are SDs significantly different (P < 0.05)?	
22		

Figure R2: PRISM – Graphpad statistical file for the PCR amplicon

Comment 3 (Figure 5C) - The new blot shown in Figure 5C still shows saturated signal in the panel with the anti-FLAG antibody (see lanes 2 and 5), so any quantification based on blots like this is problematic.

Response: Our system for ECL measurements records data in real time. So there is no issue of saturation as it happened in the past with films. We can ensure that the blots that are presented were not identified as saturated in our CCD camera-based recording system (<https://www.syngene.com/product/gbox-chemi-xx6-xx9-gel-imaging-for-fluorescence-and-chemiluminescence/>). We thank the reviewer for double checking this possible issue.

Comment 4 (Figure 5D) - To my mind, the authors' response does not clarify the point I raised.

Response: We are sorry to hear that we were not clear in our response although we tried to do our best. We all agree (authors of the study) that data provided in the revision of the manuscript appeared satisfactory to the two other reviewers. The opposite effects of Y306F and Y306F-Y316F on XBP1 mRNA splicing are very interesting and might be due to a complex pY-based regulation of RtcB functions that most probably operates not only through single tyrosine phosphorylation but also through a combination of phosphorylated tyrosine residues. This is being under current evaluation in the laboratory and for instance our most recent data indicate

that Y475 cannot be phosphorylated as a stand-alone but requires phosphorylation of at least Y306 and/or Y316. We believe that our study needs to be judged as a whole rather than by particular experiments. We presented large amount of complementary data pointing out for the same conclusions.

[Figure removed by editorial staff per authors' request]

This complex pY-based regulation of RtcB functions is also supported by our computational modeling data. Indeed, post-MD analyses of the impact of combinatorial RtcB pY on RtcB structure and pY exposure suggest that different combinations of pY yield tyrosine residues to be either "buried" within the protein globular structure or more solvent exposed. For example (see **Figure 3** and **Table S2**) the percentage of "buried residue" of Y306 changes between the two pRtcBs: in pY306, the percentage of "buried residue" is 100% against the 15% in pY306-pY316-pY475. This opposite impact on RtcB structure might reflect the reported opposite effect on XBP1 mRNA splicing of RtcB Y306F and RtcB double and triple mutants. Moreover, as part of the structural effects observed resulting from combinatorial pY on RtcB, we observed that the double (Y306, Y475) or triple (Y306, Y316, Y475) phosphorylated forms of RtcB show a N-terminus region that sticks out of the core globular region of the protein (**Figure R4**) compared to all other phosphorylated forms or to the non-phosphorylated form of RtcB. Interestingly, this region is highly positively charged and could represent a targeting signal for controlling sub-cellular localization. This is being evaluated in the laboratory at the moment (see also response to comment 6).

Figure R4: Structural impact of multiple tyrosine phosphorylations on RtcB. Only the doubly or triply phosphorylated (Y306/Y475 or Y306/Y316/Y475) forms of RtcB show a N-terminus region that sticks out of the core globular region of the protein (red circles).

Comment 5 (Figure 6G) - The new graph, which is also provided as Figure R3.2 shows that cells expressing RtcB-F wild-type and RtcB-F Y306F behave exactly the same. Nevertheless, the authors state in the text on page 12: "Both WT and Y306F RtcB-Flag rescued XBP1 mRNA splicing in PTP1B^{-/-} MEFs. Importantly, quantification of these results revealed an better rescue of the Y306F RtcB-Flag ...(Fig. 6F, G)." In my view, this contradicts the data.

Response: This reviewer is right about the graphs, but those are not corrected for the expression of Flag-RtcB constructs, a parameter that is comprised in the delta_slope graph presented in Figure 6F and that has for main objective to provide an integrated view of the effects of Y306F RtcB-Flag. As such it appears that indeed and as stated in the manuscript, the Y306F RtcB-Flag variant exhibits a better rescue than the wild-type RtcB-Flag. **Here we submit a revised version of the manuscript including the details on how the delta-slope data were generated (on p34).** We thank this reviewer for this comment.

Comment 6 (Figure 2F) - In my opinion, it is still not clear if RtcB is directly phosphorylated by c-Abl and I agree with reviewer 2 (comment 4) that in vitro kinase assays with purified RtcB variants would need to be done to make the claim that c-Abl phosphorylates RtcB. This is particularly important as the in vitro assay shown in Figure S2D, which the authors use to claim that c-Abl directly phosphorylates RtcB, is inconclusive. The band marked with a black arrowhead in the upper panel of Figure S2D is also present in the first lane, i.e. it could reflect c-Abl autophosphorylation rather than phosphorylation of RtcB by c-Abl.

Response: We would like to provide explanations to this observations. We respectfully disagree with this reviewer comment for the following reasons:

i) the identification of phosphotyrosine-containing RtcB peptides following incubation with c-ABL was performed using mass spectrometry and therefore could be due to either a phosphorylation of recombinant RtcB by c-ABL or an

autophosphorylation of RtcB, which is not known to exhibit any tyrosine kinase activity. To test the latter point, mass spectrometry analysis of recombinant RtcB following incubation in the same conditions but without the c-ABL kinase was also achieved and mass spectrometry analysis did not identify any phosphotyrosine containing peptide, thus indicating that the phosphotyrosine (Y306, Y316 and Y475) containing RtcB peptides were a consequence of the action of c-ABL on RtcB (**Figure 2C**).

ii) As this experiment was carried out in an *in vitro* kinase assay, one can argue that these phosphorylation events could be forced because of the sole presence of a kinase and a substrate in the kinase reaction. To address this issue in a more complex system, using cultured cells, we developed phospho-specific antibodies targeting the phospho-Y475 containing peptide as this was found to be the most represented phosphopeptide of RtcB in phosphoproteomics databases and the most abundant phosphopeptide found in our mass spectrometry analysis following *in vitro* phosphorylation of recombinant RtcB by c-ABL. We demonstrated that these antibodies lost immunoreactivity against Y475F RtcB as a proof of its specificity (**Figure S3**). The use of these antibodies showed that in cells treated with either dasatinib or imatinib, both known inhibitors of c-ABL, the amounts of pY475 containing RtcB decreased (**Figure 2D, E**). This indicates that the inhibition of c-ABL in live cells lead to the attenuation of RtcB phosphorylation on a tyrosine residue found to be phosphorylated by c-ABL *in vitro*. As a final step, we used a knock-down of c-ABL in HeLa cells and evaluated the tyrosine phosphorylation of RtcB, and again in this set of experiments, attenuation of c-ABL expression led to decreased tyrosine phosphorylation of RtcB.

To sum up, our results are the following:

- In vitro kinase reaction using c-ABL as a kinase and RtcB as a substrate leads to the identification of 3 phosphotyrosine containing RtcB peptides using mass spectrometry sequencing;
- RtcB phosphopeptides containing pY475, pY306 and pY316 were identified, with the pY475 containing peptide being the most abundant;
- The phosphorylation of RtcB on Y475 is attenuated upon treatment of HeLa cells with dasatinib or with imatinib
- Total tyrosine phosphorylation of RtcB is attenuated in HeLa cells knocked down for c-ABL.

To further evaluate whether the *in vitro* kinase assay could yield specificity in the kinase reaction, we used an assay similar to that above but with SRC as kinase, since this was found to alter IRE1 signaling in cellular models (Tsai et al. PNAS 2018 115(18):E4245-E4254). The presence of phosphotyrosine containing peptides in RtcB was evaluated using mass spectrometry sequencing which now revealed only one phosphotyrosine containing peptide being present, with phosphorylation found on Y5.

[Figure removed by editorial staff per authors' request]

Altogether, our results indicate that RtcB is phosphorylated *in vitro* and in cells by c-ABL with appropriate experimental standards and also that RtcB is phosphorylated *in vitro* by SRC. The phosphopeptides identified following either c-ABL or SRC mediated phosphorylation *in vitro* might reflect the high selectivity of these kinases for specific sites on RtcB which might in turn reflect specific regulatory mechanisms. At present time the biological relevance of SRC-mediated phosphorylation is under analysis in our laboratory (see also response to comment 4).

Minor points

Comment 13 (Figure 7A) - In contrast to what the authors state in the point-by-point response, Figure 7A has not been changed at all.

Response: The reviewer is correct and this is an error in the version that was submitted. We apologize for this. The panel that was supposed to be in Figure 7A (**Figure R6**) is shown below with proper statistical comparisons, it is ready and can be replaced immediately. **Here we submit a revised version of the manuscript including this modified panel for Figure 7A.**

A

Figure R6: new panel A in Figure 7

January 25, 2022

RE: Life Science Alliance Manuscript #LSA-2022-01379-TR

Dr. Eric Chevet
INSERM U1242
CLCC eugène marquis
Rennes 35000
France

Dear Dr. Chevet,

Thank you for submitting your revised manuscript entitled "Stress-induced tyrosine phosphorylation of RtcB modulates IRE1 activity and signaling outputs.". We would be happy to publish your paper in Life Science Alliance pending final revisions necessary to meet our formatting guidelines.

- please upload your main and supplementary figures as single files
- please add the Twitter handle of your host institute/organization as well as your own or/and one of the authors in our system
- please add an Author Contributions section to your main manuscript text
- please add your main, supplementary figure, and table legends to the main manuscript text after the references section
- please add callouts for Figures S3A-C, S4A-B, and S5A-B to your main manuscript text
- please add a Data Availability Statement to highlight that deposition of the mass spec data

FIGURE CHECKS:

- please make sure that the Source Data is clearly labeled so that Readers can easily identify which Figure it should match
- In your response to Referee #3 Comment 2 about Figure 5A, please provide the raw numbers, not just the statistical analysis that you've already provided. This can be provided in a final rebuttal letter.
- the quality of all figures should be at least 300 dpi
- In your response to Referee #3 Comment 1 about Figure 1E, you mention that you addressed the t-test questions in the figure legend and highlighted them in blue. I do not see this, please update (highlighting is not necessary, but the information is).

A. FINAL FILES:

B. MANUSCRIPT ORGANIZATION AND FORMATTING:

Sincerely,

January 31, 2022

RE: Life Science Alliance Manuscript #LSA-2022-01379-TRR

Dr. Eric Chevet
Inserm
INSERM U1242
CLCC eugène marquis
Rennes 35000
France

Dear Dr. Chevet,

Thank you for submitting your Research Article entitled "Stress-induced tyrosine phosphorylation of RtcB modulates IRE1 activity and signaling outputs.". It is a pleasure to let you know that your manuscript is now accepted for publication in Life Science Alliance. Congratulations on this interesting work.

DISTRIBUTION OF MATERIALS:

Again, congratulations on a very nice paper. I hope you found the review process to be constructive and are pleased with how the manuscript was handled editorially. We look forward to future exciting submissions from your lab.

Sincerely,
